

**Odds and ends of atmospheric mercury in Europe and over northern Atlantic Ocean:**
**Temporal trends of 25 years of measurements.**
Danilo Custódio[1]*, Katrine Aspmo Pfaffhuber[2], T. Gerard Spain[3], Fidel F. Pankratov[4], Iana
Strigunova[5, a], Henrik Skov[6], Koketso Molepo[1], Johannes Bieser[1], Ralf Ebinghaus[1].
[1] Helmholtz-Zentrum Hereon, Institute of Coastal Research, Max-Planck-Str. 1, D-21502 Geesthacht, Germany.
[2]NILU – Norwegian Institute for Air Research, Kjeller, Norway.
[3] National University of Ireland, Galway, Ireland.
[4]Institute of Northern Environmental Problems, Kola Science Center, Russian Academy of Sciences, Fersman Str.
14A, Apatity, 184200, Russia.
[5] Meteorological Institute, MI, Universität Hamburg, Hamburg, Germany.
[6]Department of Environmental Science, iClimate, Aarhus University, Frederiksborgvej 399, 4000 Roskilde,
Denmark
[a]International Max Planck Research School on Earth System Modelling, Hamburg, Germany.
* Correspondence to: Danilo Custodio (danilo.custodio@hereon.de)
Manuscript aim:
To determine the atmospheric mercury trend on a continental scale and evaluate
the driving factor of the downward trend in mercury in the Northern Atlantic and
Europe. Also, to assess the time variability in the light of atmospheric transport
patterns, and regional sources.
**Abstract**
The Global Monitoring Plan of the Minamata Convention on Mercury was established to
generate long-term data necessary for evaluating the effectiveness of regulatory measures
at a global scale. After 25 years monitoring (since 1995), Mace Head is one of the
atmospheric monitoring stations with the longest mercury record, and has produced
sufficient data for the analysis of temporal trends of Total Gaseous Mercury (TGM) in

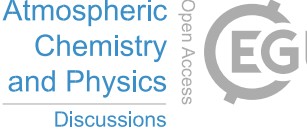

Europe and the Northern Atlantic.  Using concentration-weighted trajectories for
atmospheric mercury measured at Mace Head as well as other five locations in Europe,
Amderma, Andøya, Villum, Waldhof and Zeppelin we identify the regional probabilistic
source contribution factors and their change for the period of 1996 to 2019.
Temporal trends indicate that concentrations of mercury in the atmosphere in Europe and
the Northern Atlantic have declined significantly over the past 25 years, at a non-monotonic
rate averaging of 0.03 ng m$^{-3}$ year$^{-1}$. Concentrations of gaseous mercury at remote marine
sites were shown to be affected by continental long-range transport, and evaluation of
reanalysis back-trajectories display a significant decrease of atmospheric mercury in
continental air masses from Europe in the last two decades. In addition, using the
relationship between mercury and other atmospheric trace gases that could serve as a
source signature, we perform factorization regression analysis, based on positive rotatable
factorization to solve probabilistic mass function. We reconstructed atmospheric mercury
concentration and assessed the contribution of the major natural and anthropogenic
sources.  The results reveals that the observed downward trend in the atmospheric mercury
is mainly associated with a factor with a high load of long-lived anthropogenic species.


## 1 Introduction

Mercury is a toxic pollutant of crucial concern to public health globally. Due to its
neurotoxicity, bioaccumulation, and long-range atmospheric transport, mercury was added
to the priority list of several international agreements and conventions dealing with
environmental protection, including the Minamata Convention on Mercury (e.g. Driscoll et
al., 2013). Following the entry-into-force of the Stockholm Convention (SC) in 2004
accompanied by the Minamata convention in 2013 to restrict releases of mercury and its
compounds to the environment, a Global Monitoring Plan was devised to evaluate the
effectiveness of regulatory measures at regional and global scales. At this time, regions such
as Western Europe and North America have already established monitoring networks for

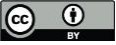



mercury in air and precipitation some of which have been in operation since the 1990s
(Schmeltz et al., 2011; Gay et al., 2013; EMEP, 2020; www.gmos.eu; www.gos4m.org).

During the past decades, atmospheric mercury concentrations in the Northern Hemisphere
decreased substantially (Slemr et al., 2003; Cole et al., 2014; Steffen et al., 2015; Weigelt et
al., 2015; Weiss-Penzias et al. 2016; Marumoto et al., 2019; Custodio et al. 2020). This
downward trend has been attributed to decreasing emissions from the North Atlantic Ocean
due to decreasing mercury concentrations in subsurface water (Soerensen et al., 2012) and
more recently to decreasing global anthropogenic emissions mainly due to the decline of
mercury release from commercial products (Horowitz et al., 2014) and the changes of
$Hg^0/Hg^{2+}$ speciation in flue gas of coal-fired utilities after implementation of NOx and $SO_2$
emission controls (Zhang et al., 2016). Mercury uptake by terrestrial vegetation has also
been recently proposed as a contributor to the downward trend (Jiskra et al., 2018).
As reported by Lyman et al. (2020), the mercury emission to the atmosphere is continuously
changing. Its monitoring is needed to track the trends, identify persistent and new sources,
and assess the efficacy of mercury pollution control policies.
In a 5-year source apportionment study, Custodio et al. (2020) show that a factor with high
load of long lived anthropogenic atmospheric species could explain the decrease of total
gaseous mercury (TGM) at Mace Head. This decrease is consistent with a decrease in the
anthropogenic mercury emissions inventory in Europe and North America (Horowitz et al.
2014). Wu et al., (2016) estimated that China's emissions also decreased since 2012 which
could have a hemispheric effect.  However, the downward trend of global anthropogenic
mercury emissions needs to be confirmed by atmospheric observations, and a long-term
evaluation of the time series of still unknown sources and its implication should be assessed.
This study reports continuous long-term temporal trends of gaseous mercury in the
Northern Atlantic, Arctic, and Europe, reporting mercury atmospheric concentrations at
Mace Head (1995-2019), Amderma 2001-2017), Andøya (2010-2019), Villum (1999-2019),
Waldhof (2005-2019), and Zeppelin (2000-2019). Here, we combine a long-time series of

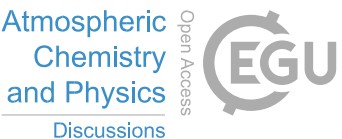

atmospheric mercury observed at these sites with calculated 120-hour reanalysis backward
trajectories in order to investigate transport and long-term changes in concentration
patterns on the regional scale.
This paper aims to evaluate the atmospheric mercury trend on a continental scale and the
emission factors contributing to the downward trend in mercury in the Northern Atlantic
and Europe.
Based on long-range Lagrangian reanalysis backward trajectories and receptor-modelling,
we investigate the trends and sources of mercury in the atmosphere, assessing the inter-
annual variability on the light of atmospheric transport patterns and changes in the regional
emissions. In addition, we exploit the observed atmospheric mercury temporal variability,
which can be used as additional constraints to improve the ability of models to predict the
cycling of mercury in the atmosphere.

**2 Experimental**

• **Sampling sites**

Data from six sites in Europe and Greenland with the longest records of atmospheric
mercury concentrations were selected for this study: Mace Head (data available 1995 –
2019), Zeppelin (2000 – 2019), Waldhof (2006 – 2019), Villum (2008 – 2019), Andøya (2010
-2019), and Amderma (2001 – 2013). Mace Head and Waldhof are mid-latitude stations,
Zeppelin, Amderma, and Villum can be classified as Arctic ones. Andøya, though at latitude
comparable to that of Amderma, is behaves more like a mid-latitude station because the
ocean around it is ice free for most of the year. At all sites mercury had been measured by
a Tekran  instrument (Tekran Inc, Toronto, Canada), more details will be given at the end of
the section



The Mace Head Global Atmosphere Watch (GAW) Station (53°20` N and 9°32W, 8 m above sea level; air-sampling inlet 18 m a.s.l.) is located on the west coast of Ireland on the shore of the North Atlantic Ocean, offering ideal conditions to evaluate both natural and anthropogenic pollutants in oceanic and continental air masses as described by Stanley et al. (2018). The station was part of the GMOS network and mercury measurements are described in detail by Weigelt et al. (2015).

The Zeppelin GAW station is located on the ridge of the Zeppelin Mountain (78°54`N, 11°52`E) at 474 m a.s.l., about 2 km from Ny Ålesund on the west coast of Spitsbergen which is the largest of the Svalbard Islands. Mercury measurements are described by Aspmo et al. (2005).

Waldhof (52°48'N, 10°45'E, 74 m a.s.l.) is a rural background site located in the northern German lowlands in a flat terrain, 100 km south-east of Hamburg., The site and analytical method are described in detail by Weigelt et al. (2013).

Villum Research Station is located at the military outpost Station Nord. It is located in the furthermost northeastern corner of Greenland on the north–south oriented peninsula of Princess Ingeborg Halvø (81°36′N, 16°40′W, 25 m a.s.l.), whose northern end is a 20 × 15 km$^2$ Arctic lowland plain. The Air Observatory is located 2 km south of the central complex of Station Nord that is manned year-round by 5 soldiers. The monitoring site is upwind of the dominant wind direction for Station Nord and thus any effect of local pollution is minimized. Atmospheric measurements at Villum are described in detail by Skov et al. (2004 and 2020).

Andøya observatory (69.3°N, 16°E, 380 m a.s.l.) is situated a few hundred meters away from ALOMAR (Arctic Lidar Observatory for Middle Atmosphere Research), which is located at



the west coast on a mountain at the island Andøya in Northern Norway. More details about
measurements at Andøya are available in Berg et al. (2008).

Amderma Polar Station is located near the Amderma settlement of the Arkhangelsk Arctic
region of Russia near the coast of the Kara Sea (69°43`N, 61°37`E, 49 m a.s.l.; Yugor
Peninsula, Russia). Gaseous mercury has been measured since 2001 until 2017. The site and
the mercury measurements are described by Pankratov et al. (2013).

At all sites mercury was measured using Tekran 2537 A and/or B instrument (Tekran Inc,
Toronto, Canada, mostly Model A, at Mace Head and Villum also Model B), an automated
dual-channel, single amalgamation, cold vapor atomic fluorescence (CVAFS) analyzer. The
instrument has two gold cartridges. While mercury is collected on one of them during the
sampling period, the other is being analyzed by thermodesorption and CVAFS detection.
The functions of the cartridges are then alternated, allowing for quasi-continuous
measurement. The instruments are usually protected by an upstream PTFE filter against
dust and aerosols and has a detection limit of ≈ 0.04 ng m$^{-3}$.
As discussed by Slemr et al. (2016), gaseous oxidized mercury (GOM) compounds are
collected on the gold cartridges and were found to be converted to elemental mercury
(GEM) probably during the thermodesorption. The instrument is thus able to measure
total gaseous mercury (TGM) provided that GOM compounds reach the cartridges. This
is frequently not the case because the GOM compounds are sticky and can thus be
removed on the way from the inlet to the cartridges (Lyman et al., 2020). The
instruments are usually protected by an upstream PTFE filter (mostly 0.2 µm, 0.4 µm at
Zeppelin, 0.45 µm at Andøya) against dust and aerosols. Additional soda-lime filters are
frequently used to remove free halogens that can shorten the lifetime of the gold
cartridges (GMOS Standard Operating Procedure, 2019) and were implemented at
Villum, Amderma, Zeppelin, and Andøya. They are suspected to capture GOM although

c Author(s) 2021. CC BY 4.0 License.



this has not been adequately tested so far (Gustin et al., 2021). Sea salt on the walls of
the sampling tubing and on the PTFE filter at coastal stations, such as Mace Head,
Andøya, Amderma, and possibly Zeppelin, is also likely to remove GOM. We conclude
that GEM is being measured at Mace Head (Weigelt et al., 2015), Villum (Skov et al.,
2020), Andøya, Amderma, and Zeppelin (Durnford et al., 2010), Waldhof (Weigelt et al.,
2013). We thus treat all data as GEM. All instruments have been operated according to
the standard operating procedures (Steffen and Schroeder, 1999; GMOS Standard
Operating Procedure, 2019).  The instruments at Villum, Zeppelin, and Andøya were run
with 5 min resolution at a sampling flow rate of 1.5 L min$^{-1}$.  At Waldhof and Mace Head
the temporal resolution was 15 min and at Amderma 30 min.
Speciated mercury measurements made at Waldhof between 2009 and 2011 provided
median concentrations of 6.3 pg m$^{-3}$ for PBM and 1.0 pg m$^{-3}$ for GOM while the median GEM
concentration was 1.6 ng m$^{-3}$, representing >99,5% of the TGM (Weigelt et al., 2013). GOM
measurements using Tekran speciation system are considered to be underestimated (Jaffe
et al., 2014; Lyman et al. 2020). Other speciation measurements show that with the
exception of polar depletion events and upper troposphere, GEM is the dominant form of
atmospheric mercury, accounting mostly for more than 95% of the TGM (Mao et al., 2016)

• **Back-Trajectory Analysis, Concentration-weighted trajectories, and probability**
**mass function models.**

To evaluate the spatial coverage and sources of air sampled at the six stations, three
dimensional reanalysis air mass back-trajectories at an arrival height of 50 m and 500 m
above ground were calculated at each site for 120 h using HYSPLIT (v.4.2.0,
NOAAhttps://www.arl.noaa.gov/hysplit/hysplit/). Stein et al. (2015) describes the method
of choosing the trajectory heights and time initiations). Two trajectories were calculated
per day, each representing an average trajectory for the period of 12 h. All individual back-
trajectories generated by HYSPLIT were converted to text shape files and imported into R



(R Project for Statistical Computing), merged with concentration files and used for spatial
analysis. To account for the speed and atmospheric residence time of air masses, each
continuous back-trajectory line was transformed into 120 hourly points.
Concentration-weighted trajectories (CWT), is an approach which can be used to indicate
the probability of a grid cells contribution to pollution events (Cheng at al. 2013). It is based
on a statistical model and can incorporate meteorological information in its analysis scheme
to identify the average concentration in areas for pollutants based on a conditional
probability that an air parcel that passed through a cell with a gradient concentration
displays a high concentration at the trajectory endpoint (Ashbaugh et al. 1985, Byčenkienė,
et al. 2014). The CWT obtained at this study     are a function of average mercury
concentrations that were obtained every 12 h and of the residence time of a trajectory in
each grid cell. The 12-hour trajectory segment endpoints for each back trajectory that
corresponds to each 12 h TGM, or GEM, were retained. For a 120-hour trajectory duration,
84 trajectory segment end points were calculated. This transformation of trajectories into
hourly segments allowed the subsequent application of a kernel density tool to the
combined back-trajectory air mass points from all sampling sites in order to create a density
map of the continental concentration and spatial coverage of concentration airflows
sampled at the sampling site over the course of an entire year. Seasonal back-trajectory
maps were also generated for evaluation of potential seasonal changes in the coverage and
sources of airflows (with seasons defined as summer (June, July, and August), autumn
(September, October, and November), winter (December, January, and February), and
spring (March, April, and May).In this study, the assessment of CWT was performed on
annual bases, the concentrations in grid cells were calculated by counting the average
concentration of trajectory segment end points that terminate within each cell as described
by Byčenkienė, et al. (2014) and Tang et al. (2018).
The source apportionment for Mace Head was performed based on the mass conservation
principle with the inclusion of potential rotated infinity matrices transformation producing
factors with chemical profile signed by tracer species linked to its source. The full description

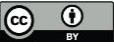

of PMF and its reconstruction consideration, chemical species considered, uncertainties,
and constraining of factors are presented in Custodio et al. (2020). In this study, the PMF
was applied to the Mace Head daily data. The species considered in the factorization and
their mass loaded in each factor are displayed in Figure 3S in the article supplement section.
In addition, the reconstructed GEM and the observation displayed an $r^2$ of 0.9949. The GEM
mass solved by factorization agree into the 10/90th percentile quantile regression, as shown
in Figure 4S in the supplement of the article.

**3 Results and discussion**

In this section, we present the time series and trends of GEM concentrations from a data
set covering the period from February 1996 to December 2019 (Mace Head), July 2001
to March 2017 (Amderma), February 2000 to December 2019 (Zeppelin), January 2006
to December 2019 (Waldhof), from January 2004 to December 2019 (Andøya), and from
June 1999 to December 2019 (Villum). At Villum the measurements covered only 6
months (spring, summer, and early autumn) in 1999 – 2002, and no measurements are
available for the years 2003 – 2008 (Skov et al., 2020). The data are summarized in in
Figure 1.
GEM concentrations and their frequency distributions shown in Figure 1 display distinct
differences between the stations. GEM concentrations at Villum, Amderma, and
Zeppelin decrease frequently to values near zero (minimaof 0.0, 0.0, and 0.1 ng m$^{-3}$ at
Villum, Amderma, and Zeppelin, respectively) and their frequency distribution is skewed
to lower values as documented by somewhat lower average than median GEM
concentrations and the lowest 5[th] percentiles of all sites with 0.55, 0.62, and 1.04 ng m$^{-}$
$^3$ at Villum, Amderma, and Zeppelin, respectively. The seasonal occurrence of the polar
depletion events at these three stations is characteristic for the Arctic sites with ice and
snow coverage (Steffen et al., 2008). The GEM frequency distribution at Zeppelin is less

c Author(s) 2021. CC BY 4.0 License.

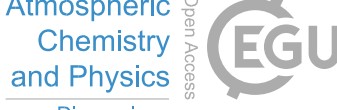

skewed than at Villum and Amderma perhaps because of the Zeppelin altitude of almost
500 m asl, which is above the layer with most intensive halogen chemistry within the
first 100 – 200 m above snow (Tackett et al., 2007).
The distribution of GEM concentrations at Waldhof, a mid-latitude station in Central
Europe, is on the contrary skewed to higher values because of frequent events with local
and regional pollution (Weigelt et al., 2013). The average and median GEM
concentrations at Waldhof are the highest of all the investigated stations, and the
average GEM concentration is substantially higher than median one.
The frequency distribution at Andøya is nearly symmetric, neither skewed to low nor to high
GEM concentrations although a pronounced seasonal variation can be observed. At latitude
comparable to that of Amderma there are no pronounced depletion events at Andøya
because it is exposed to Gulf stream and as such free of ice for most of the year. Events with
local and regional pollution are also largely missing at Andøya (95th percentile of 1.79 ng m⁻
$^3$ is lower when compared with 2.32 and 2.96 ng m$^{-3}$ at Waldhof and Mace Head). GEM
frequency distribution at Mace Head is similar to that at Andøya and the average and
median GEM concentrations are nearly the same as both stations are exposed to air
originating mostly from the Atlantic Ocean. Opposite to Andøya, GEM frequency
distribution at Mace Head is slightly skewed to higher concentration because of the local
pollution and occasional air transport from Europe (Weigelt et al., 2015).

**3.1 Seasonal variation**
Figure 2 shows similar seasonal variations at Mace Head, Waldhof, and Andøya with the
maximum GEM concentrations in late winter and early spring and the mimimum ones in
late summer and early autumn. Similar seasonal variation has been observed at most of the
mid-latitude sites in the northern hemisphere (e.g. Cole et al., 2014; Weigelt et al., 2015;
Sprovieri et al, 2016, Angot et al., 2016). It is usually accompanied by a summer maximum
in wet deposition (Gratz et al., 2009; Prestbo and Gay, 2009; Zhang and Jaeglé, 2013;
Sprovieri et al., 2017) which is caused by faster oxidation of Hg$^0$ to Hg$^{2+}$ in summer providing

c Author(s) 2021. CC BY 4.0 License.



more Hg$^{2+}$ for scavenging by rain (Holmes et al., 2010; Zhang et al., 2012; Zhang and Jaeglé,
2013; Horowitz et al., 2017). GEM uptake by vegetation can also contribute to summer
minimum of GEM concentrations at midlatitudes (Jiskra et al., 2018).
Seasonal variations in mercury at Amderma, Villum and Zeppelin are influenced by polar
depletion events in spring and the subsequent reemission of the deposited mercury from
snow in summer which result in pronounced GEM minima in April and May and maxima in
July (Steffen et al., 2008, 2015; Dommergue et al., 2010; Cole and Steffen, 2010; Cole et al.
2014; Angot et al., 2016; Skov et al. 2020). A similar pattern is also observed at Alert (Cole
et al. 2014). Note the larger amplitude of seasonal variation at Arctic stations (0.8 – 1.2 ng
m$^{-3}$) when compared to the mid-latitude ones (0.95 – 1.07 ng m$^{-3}$). Zeppelin has a
substantially smaller amplitude of seasonal variation than Amderma and Villum, probably
because of its altitude as already noted in the discussion of the frequency distributions.
Andøya, although located at a comparable latitude as Amderma, is only slightly influenced
by the polar depletion events because it is ice-free for most of the year, as already
mentioned.
Figure 2 shows density maps which are based on the seasonal mean mercury concentration
associated with respective trajectories which arrived synchronously at all six stations.  The
northern parts of the spring and summer panels show over the Arctic Ocean the lowest and
highest mercury concentrations, respectively, which is consistent with the spring polar
mercury depletion and summer emission of the mercury deposited during the depletion
events. The highest GEM concentrations over the middle of the North Atlantic occur in
winter, the lowest ones in summer and autumn which is consistent with the seasonal
variations at Mace Head and Andøya. High GEM levels over large part of the Europe occur
in all seasons. The highest concentrations by level and extension occur in winter and spring,
somewhat lower in summer and autumn.

**3.2** Temporal trends and regional source of TGM

c Author(s) 2021. CC BY 4.0 License.

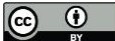



Figures 3 and 1S show the Kernel-regression of mercury concentrations at Mace Head,
Amderma, Andøya, Villum, Waldhof, and Zeppelin. Both figures show a non-monotonic
concentration change with temporary increases to intermediate maxima at Waldhof,
Zeppelin, and most pronounced at Villum with a maximum in 2013. The overall trend of
GEM concentrations at all sites points in downward direction. Table 1 summarizes the
overall trends calculated by least-square-fit (LSQF) from monthly medians and compares
them with those at Mace Head over the same periods of available measurements. Averages
of monthly medians over the same periods are also listed. Monthly medians were chosen
to reduce the influence of depletion events at polar stations and of pollution events at
midlatitude stations. Mace Head was taken as a bench mark because of the longest and
most complete data record. In addition, the trend at Mace Head represents the baseline
trend (Weigelt et al., 2015). All trends in the table are significant at >99.9% level as are the
differences between the trends at the sites and those at Mace Head.
GEM concentration at Mace Head decreased with an annual rate of $-0.0244 \pm 0.0011$ ng m$^{-3}$
yr$^{-1}$ in 25 years ($-0.0256 \pm 0.0012$ ng m$^{-3}$ yr$^{-1}$ in 24 years). For different periods within these
long-term measurements, the decrease rate at Mace Head varied between $-0.0244$ and $-$
$0.0346$ ng m$^{-3}$ yr$^{-1}$ as illustrated by Figure 3. The average GEM concentrations at Waldhof
are substantially higher than those at Mace Head demonstrating the continuing presence
of regional emissions. The downward trend at Andøya is comparable to that at Waldhof but
substantially smaller than at Mace Head for the period of Andøya measurements. The
average GEM concentration at Andøya is somewhat higher than at Mace Head.
Of the Arctic stations, GEM concentration at Zeppelin decreased with only $-0.0087$ ng m$^{-3}$
yr$^{-1}$ when compared to $-0.279$ ng m$^{-3}$ yr$^{-1}$ for the same period at Mace Head. Cole et al.
(2013) have reported a trend of $+0.002$ ng m$^{-3}$ yr$^{-1}$ ($-0.007$ to $+ 0.012$ ng m$^{-3}$ yr$^{-1}$, 95%
confidence range) for Zeppelin in the decade 2000 – 2009 which is consistent with the trend
value presented here for 2000 – 2019.  The average GEM concentration of $1.57 \pm 0.24$ ng m$^{-3}$
for the decade 2000 – 2009 (Cole et al., 2013) is almost identical with $1.55 \pm 0.14$ ng m$^{-3}$
reported here for the years 2000 – 2019, too. A somewhat higher but comparable decrease



rate of -0.012 ng m$^{-3}$ yr$^{-1}$ (-0.021 to 0.000 ng m$^{-3}$ yr$^{-1}$, 95% confidence interval) was reported
for Alert for the 2000 to 2009 period (Cole et al., 2013). The average GEM concentration of
1.50 ± 0.35 ng m-3 at Alert is also comparable to that of Zeppelin in the 2000 – 2009 period
(Cole et al., 2013). Figure 3 shows at Zeppelin a broad maximum around 2006.
Based on LSQF the GEM at the Arctic stations Amderma and Villum behave differently. The
downward trends of -0.0327 ± 0.0047 and -0.0409 ± 0.0072 ng m$^{-3}$ yr$^{-1}$ at Amderma and
Villum, respectively, are roughly comparable and both are substantially larger than those at
Mace Head for the respective periods. Their trend uncertainties are substantially larger than
the uncertainties at the other stations. On the other side, the average GEM concentrations
at Amderma and Villum are comparable to those at Mace Head for the respective periods,
albeit with substantially higher standard deviations. This is partly due to the short periods
with varying trend at Amderma and even a pronounced temporal maximum at Villum. The
downward trend at Amderma can be potentially enhanced by moving the station in 2005
and 2011. As reported by Pankratov et al. (2013) gaseous mercury at Amderma can variate
depending on the distance to the coast, with effect mainly the incidence of extremes events.
However, downward concentrations at this station was observed even at the segments
before 2005 and afterword when the station was moved 10 km to the coast side.
The higher level of atmospheric mercury at Villum in 2013 is consistent with an elevated
mercury level over Greenland in that year, as deduced from backward trajectory analyses
shown in Figure 4. Large subglacial source of mercury at Greenland has been recently
reported by Hawkings et al. (2021). The increase of GEM at Villum in 2010 and 2013, which
drives the trend up during this period, corresponds to two periods of negative extreme at
Arctic Oscillation (AO). The extreme on AO and North Atlantic Oscillation (NAO) can enhance
the mercury discharge from ice to the atmosphere. Bevis et al. (2019) report an anomalous
ice mass loss at Greenland in the 2010-2014 epoch. The abrupt ice melting was driven
mainly by changes in air temperature and solar radiation caused by atmospheric circulation
anomalies.

c Author(s) 2021. CC BY 4.0 License.

In addition, the negative phase of the summertime NAO index increases the prevalence of
high pressure, clear-sky conditions, enhancing surface absorption of solar radiation and
decreasing snowfall, and it causes the advection of warm air from southern latitudes into
Greenland. These changes promote higher air temperatures, a more extended ablation
season and enhanced melt ice (Fettweis et al. 2013). In 2014/2015, when the AO indexes
again turned positive and NAO negative, significant ice loss was reestablished (Bevis et al.,

2019).

The back trajectories of air masses calculated for each site were combined with the
measured concentration at a 12h time resolution. The results were used to identify possible
regional sources and also to assess temporal variations. Figure 4 shows that calculated air
mass back-trajectories for the five monitoring sites mainly reflect air masses transported
from the ocean, however, they also indicated elevated concentrations in continental
trajectories such as from central Europe which are due to anthropogenic emission sources.
Despite a shift to the south that can be associated with uncertainties in the Lagrangian
approach, the airflow patterns and concentrations hotspot were consistent with the current
knowledge of geolocation of GEM sources in Europe (Panagos et al. 2021). Figure 4 also
shows a high level of mercury associated with air masses coming from the northwest
(Canada and Greenland) during the 1997-2000 epoch, 2005, 2010, 2014 besides of 2013
already mentioned.
The most revealing detail in the observed trend of GEM is displayed in Figure 4, where it is
noticeable that the downward trend is ongoing on a regional scale. This decrease could
represent a change in the balance between sources and sinks of mercury in the atmosphere.
The downward trend seems to be driven by decreasing concentrations in continental
Europe. This phenomenon is observed mainly after 2005 when data from Waldhof is
considered. The downward trend in mercury concentration is observed in all trajectories,
even in remote areas, indicated by the yellow fades to green. This phenomenon can be
explained only by reductions in global atmospheric mercury sources. In addition, Figure 4

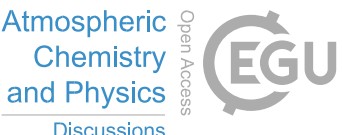

also shows that the decrease is more pronounced in the hotspot areas identified as
anthropogenic sources, where the colour shifts from dark to light red in plots from 2005 to

2019.

The later downward trend at Zeppelin and Villum (Figure 3, 1S), suggests that these remote,
high latitude stations are less affected by direct European continental emission.
An accurate emissions inventory is essential for interpreting trends in atmospheric
concentrations and assessing the effectiveness of mercury pollution control policies (Lyman et
al., 2020). However, the observed GEM trend is not consistent with the global anthropogenic
emissions inventory, in which uncertainties ranged from -33% to 60% (Lyman et al. 2020 and
references in).
The seemingly non-monotonic downward trends with inter-annual ups and downs observed
in this study are not well explained. However, an inspection of the Mace Head data (e.g. in
Figure 3 and 4) reveal that this trend is composed of two segments: one starting in 1999
and ending approximately in 2010 and a second one in 2014 after a biennial upward
tendency. It could be premature to assume that the atmospheric mercury trend can be
driven simply by a political decision. However, important mercury control and regulation
have been taken since later 90`s which result into the mercury international treaty
(Minamata Convention on Mercury) designed to protect human health and the
environment from anthropogenic emissions and releases of mercury approved on 10
October, 2013. Continental and international environmental treaties are the result of long
political and societal debate and commitment to such deal could reflect an already
established control policy at the national level.
For example, in 1990 The United States Clean Air Act, put mercury on a list of toxic pollutants
that needed to be controlled to the greatest possible extent, forcing industries that release
high concentrations of mercury into the environment to install maximum achievable control
technologies (MACT). In 2005, the EPA promulgated a regulation that added power plants
to the list of sources that should be controlled and instituted in the nation, and in 2011 new

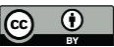



rules for coal-fired power plants were announced by EPA (State of new Jersey, at al. 2008,
*Castro Mark S., Sherwell John 2015*).
Additionally, in 2007 the European Union implemented new mercury control measures,
banning mercury in new non-electrical measuring devices, such as thermometers and
barometers (Jones, H. 2007).
We note that Waldhof, a continental station close to anthropogenic sources in Europe,
corroborates the interpretation of an anthropogenic emission driven mercury trend. This
station shows a more pronounced GEM decrease between 2005-2010 compared to the
years since then. Zhang et al. (2016) presented a revised inventory of Hg emissions for the
estimation of artisanal and small-scale gold mining emissions, and, accounting for the
change in $Hg^0/Hg^{II}$ speciation of emissions from coal-fired utilities after implementation of
emission controls targeted at $SO_2$ and $NO_x$, those authors estimate a factor of 20% decrease
in atmospheric emission from 1990 to 2010. Natural sources can contribute up to 40% of
the atmospheric mercury budget (Pirrone, et al., 2010); however, a trend on such a source
is not observed or reported in the literature, so far.
Based on the GEM associated with each air mass trajectory, we investigated the impact of
atmospheric circulation on continental Europe and Northern Atlantic Ocean and observe
distinct concentration patterns for the ocean and continental regions.  We observed for
example, that air masses arriving at Mace Head from central Europe show distinct trends.
We compared the regional patterns of GEM with other pollutants (CO, $CO_2$, $CH_4$, $O_3$, $CHCl_3$,
$CCl_4$, and CFCs) also measured at Mace Head and find that GEM shows a similar pattern
concerning source location as the other species closely related to anthropogenic sources.
However, GEM displays a downward trend, with decreasing concentrations in air masses
from central Europe and England.
Figure 4 shows the concentration- weighted trajectory maps for GEM measurements over
Mace Head, Amderma, Andøya, Villum, Waldhof and Zeppelin. It can be seen that the
highest concentrations are almost exclusively from air masses over central Europe.



Exceptions are 1997 to 2000 which indicate high levels of GEM in air masses coming from
Northwest. However, it should be mentioned that CWT for this period computed only Mace
Head data and Villum (1999-2000).
The results also show a lower level of GEM in air masses segments over the North Atlantic
region. This region is constantly associated with a sink of anthropogenic pollutants.

**3.3 Probability of source contribution.**
Based on our analysis so far, our hypothesis is that the mercury concentration in North
Atlantic air masses is affected by the intensity of transport from important regional and
global sources and also by temporal changes in these sources. For example, the high
mercury concentrations observed in the late 1990`s coincide with higher contributions from
continental air masses. During 2001, a noticeable reduction in the Mace Head GEM
concentration was observed, corresponding to a lesser influence of continental European
air masses. This was due not only to a lower frequency of air masses from continental
Europe but also lower concentration of GEM in those air masses compared to previous
years. A similar phenomenon was observed in the trend during 2005/2006 and 2008 to 2010
when an increase and decrease of inter-annual trend corresponded to higher and lower
CWT in air masses coming from continental Europe (Figure 2S).
In a five-year source apportionment of mercury at Mace Head, Custodio et al. (2020) show
that a factor with high load of anthropogenic species could explain downward trends of
GEM. The downward trend of that factor was associated with a reduction in emissions due
to cleaner manufacturing processes involving mercury and regulations limiting the
emissions from coal-fired power plants since the 1980s, as well as a reduction in the release
of mercury from commercial products since 1990s (Streets et al. 2011, Zhang et al., 2016).
Here we extend the source apportionment analysis back to 1996. The extended
reconstruction of the main sources of mercury back to 1996, shown in Figure 5, displays a



similar apportionment pattern to that reported by Custodio et al. (2020). The source
apportionment indicates a baseline factor characterized by high load of anthropogenic
species accounting for 65% of GEM mass. The baseline factor has already been proposed as
the driving factor for mercury trends at Mace Head by Custodio et al. (2020). In this study,
this factor displays a downward trend of 2.7 % yr$^{-1}$, and correlates (r =0.97) with the mercury
trend (Figure 6). A factor with load of anthropogenic species driving the Mace Head GEM
trend down by a strength of 97 % at the level of 0.001 (p-values) is also supported by Figure
4, which displays a temporal decrease in mercury level in reanalysis backward trajectory.
One important consideration to take into account is that the baseline factor is interpreted
as global mercury budget from several sources which were not solved by PMF, such factor
could also take into account the strength of non-modulated extremes events or periodic
oscillations such as ENSO as speculated by Slemr et al. (2020) and references therein, those
events can be a reason for increase rotation in the mercury trend, imposing significance and
raising the correlation.
The Global Mercury Assessment inventory (AMAP/UNEP, 2019) estimates a contribution of
combustion sources to atmospheric mercury at 24%. In this study the combustion factor,
which was indicated by high load of CO, accounted for 20% of total GEM mass at Mace Head
(Figure 5). A slight decreasing trend was observed in this factor, which could be associated
with the implementation of emission controls on coal-fired utilities as proposed by Zhang
at al. (2016) in a revised inventory of Hg emissions.
However, as reported by Custodio at al. (2020) this trend should be taken with caution since
the combustion factor was fingerprinted by CO, a short-lived species (1-3 months) with
significant seasonal and atmospheric transport dependence.
The ocean factors account for 12% of total GEM mass at Mace Head and was identified by
a high load of CHCl$_3$ (Figure 5). CHCl$_3$ used to trace sign ocean factor, is a trace atmospheric
gas originating 90% from a natural source, being offshore seawater the largest issuer
(McCulloch, 2003).



As reported by Custodio et al. (2020) and references therein, the residence time of mercury
in the ocean is substantially longer than in the atmosphere, ranging from years to decades
or millennia. Human activity has substantially increased the oceanic mercury reservoir and
consequently is affecting the fluxes of mercury between the sea and atmosphere (Strode et
al., 2007).
The acidification of oceans, climate change, excess nutrient inputs, and pollution are
fundamentally changing the ocean's biogeochemistry (Doney, 2010) and will certainly also
influence mercury ocean-air fluxes in a still unknown direction.
This study shows an upward trend in the oceanic factor after 2010, as can be seen in Figure
(5), however its significance, implication and causes remain to be determined.

**4 Conclusion**

A conundrum in the observed negative trend in mercury in Europe and Northern Atlantic
over the past two decades is explained in this study by a decrease in anthropogenic
emissions. The significant decline in concentrations of GEM over the past two decades
demonstrates that regulatory measures across Europe have been successful in reducing the
atmospheric concentration of this species although an extensive fossil fuel use and a legacy
of stockpiles in the environment continue to pose a challenge.
These results show the transport pattern of atmospheric mercury and reveal that a baseline
factor with a high load of long-lived anthropogenic species dominates the source of mercury
in the Northern Atlantic and highlight the need for continued monitoring of the gaseous
mercury and its sources. This study brings a monitoring concept for mercury on a
continental scale which can be extended to a Global Monitoring plan by integration of   the
mercury monitoring network, potentially identifying hotspot concentration areas and their
change over time.



This large-scale, long-term trend data evaluation can be used for assessing the effectiveness
of the Minamata Convention.
More specific conclusions include the following:
➢ Enhancement of mercury in the air masses over Greenland in summer during

epochs of atmospheric circulation anomalies.

➢  Mercury downward trends of $2 \pm 3\%$ yr$^{-1}$, $2.1 \pm 1.5$ % yr$^{-1}$, $1.6 \pm 3.9$ % yr$^{-1}$, $4 \pm 16$

527        % yr$^{-1}$, $2 \pm 4$ % yr$^{-1}$, and $3 \pm 3$ % yr$^{-1}$ at Amaderma, Andøya, Mace Head, Villum,

Waldhof  and Zeppelin respectively are influenced by regional sources and then

biased for global trend.

➢ The observed GEM downward trend at Northern Atlantic and Arctic seems to be

driven by decreasing in concentration in continental Europe.

➢ A baseline factor with high load of anthropogenic species drives the mercury trend

down by a strength of 97 % at the level of 0.001 (p-values) based on source

reconstruction at Mace Head.

➢ Combustion sources could account for 20 % of GEM with a slightly decreasing

trend, and ocean sources account for 12 % with a slightly increasing trend.


**Authors Contribution:**
**DC** proposed the article, processed data and wrote the article. **KAP** provided data and evaluated the findings.
**TGS** provided data, support the writing and discussions. **FFP** provided data and participate in the discussion. **IS**
supported the calculation in scripts, data assimilation, besides provide meteorological and Lagrangian analysis.
**K**P supported the trajectories calculation and discussion. **HS** provided data and participate in the discussion. **JB**
and **RE** endorse and supported the article preparation, respectively.


Table 1: Comparison of GEM trends and average concentrations at Zeppelin, Waldhof, Andøya,
Amderma, and Villum with those at Mace Head. The trends (± confidence interval at 95% level) were
calculated by the least square fit (LSQF) of monthly medians over the same months for which the
measurements are available. Average GEM concentrations were calculated as average of monthly
medians over months with synchronous measurements.

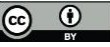


| Site | Period, number of months | Trend [ng m⁻³ yr⁻¹] | | GEM average concentration [ng m⁻³] | |
|---|---|---|---|---|---|
| | | Site | Mace Head | Site | Mace Head |
| Mace Head | Feb 1996 – Dec 2020, 279 | | -0.0244 ± 0.0011 | | |
| Mace Head | Feb 1996 – Dec 2019, 267 | | -0.0256 ± 0.0012 | | |
| Zeppelin | Feb 2000 – Dec 2019, 222 | -0.0087±0.0015 | -0.0279 ± 0.0013 | 1.548±0.141 | 1.483±0.196 |
| Waldhof | Jan 2006 – Dec 2019, 161 | -0.0243±0.0025 | -0.0280 ± 0.0022 | 1.649±0.161 | 1.399±0.158 |
| Andøya | Jan 2004 – Dec 2019, 119 | -0.0262±0.0023 | -0.0346 ± 0.0029 | 1.519±0.127 | 1.368±0.165 |
| Amderma | Jul 2001 – Mar 2017, 133 | -0.0327±0.0047 | -0.0257 ± 0.0022 | 1.480±0.265 | 1.517±0.153 |
| Villum | Sep 2008 – Jun 2019, 111 | -0.0409±0.0072 | -0.0293 ± 0.0031 | 1.372±0.274 | 1.371±0.140 |



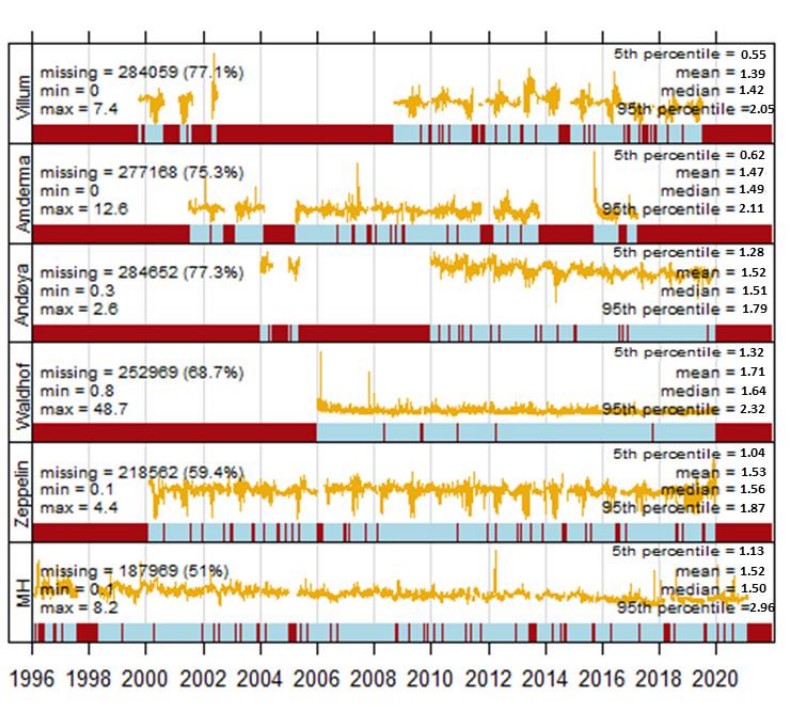

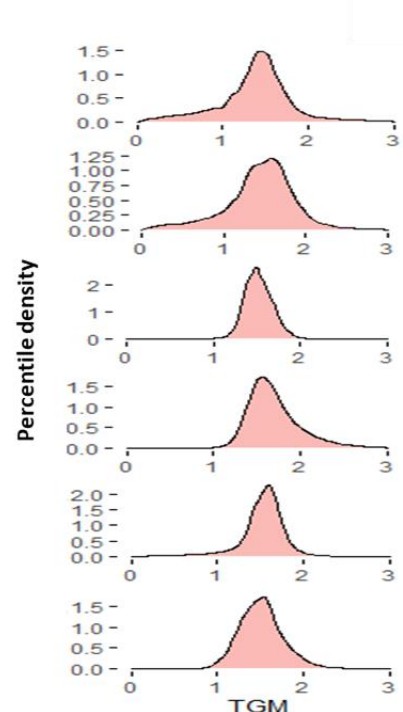




Figure 1: Summary of time series of GEM (ng m$^{-3}$) in hourly resolution measured at Mace
Head, Zeppelin, Waldhof, Andøya, Amderma and Villum on the left side. Distributions
density of the measured concentrations on the left side. *The red and blue bars on the time
axis represent the missing and available data periods, respectively.



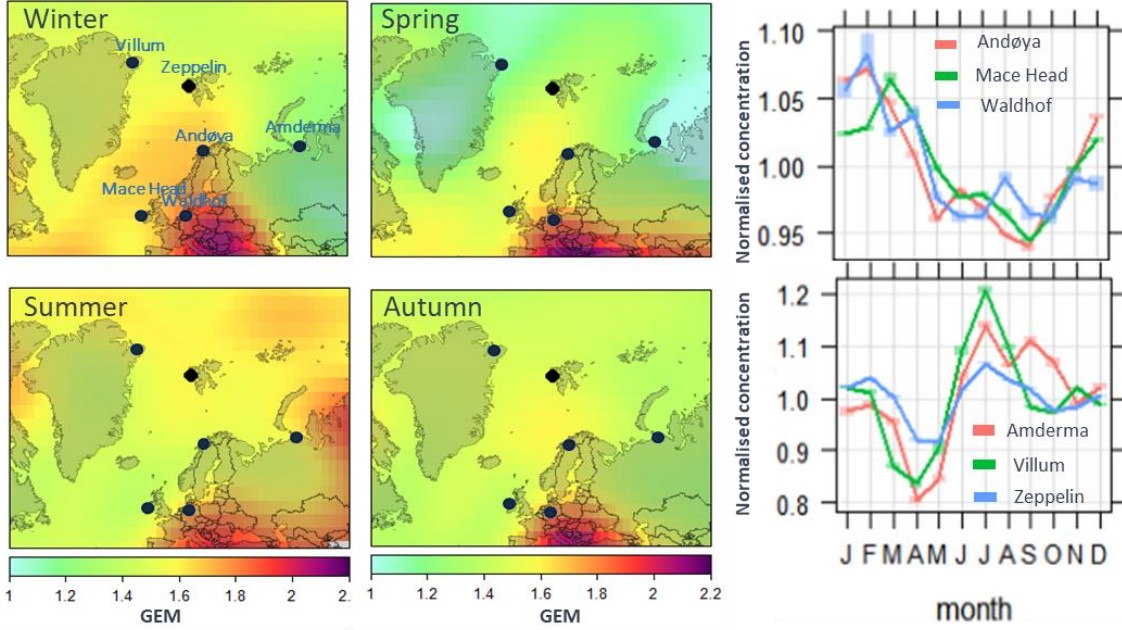


Figure 2: Left panels: The density map of atmospheric mercury concentrations in different seasons.
Right panels: Normalized annual variation of the mercury concentrations at Arctic stations
(Amderma, Villum, Zeppelin) and at the mid-latitude ones (Mace Head, Waldhof, and Andøya). The
shaded areas are the 95% confidence intervals for the monthly mean.


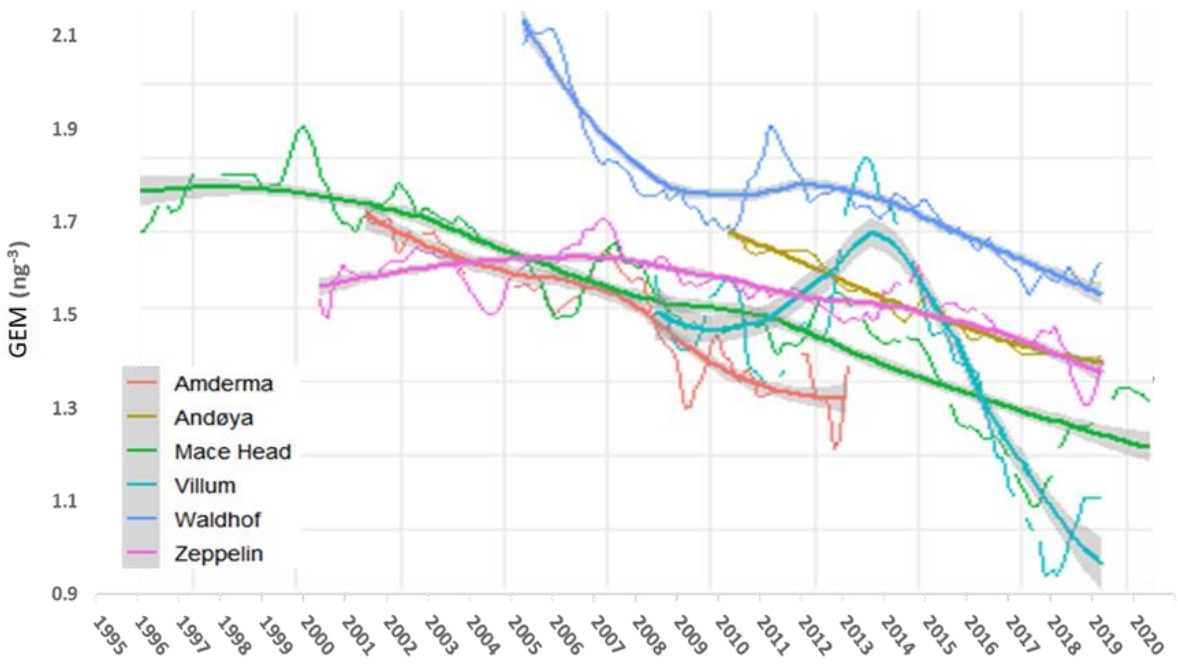


Figure 3. Kernel-regression of GEM at Amderma, Andøya, Mace Head, Villum, Waldhof, and Zeppelin
for the period of 2001-2013, 2010-2019, 1995-2019, 2008-2019, 2006-2019, and 2000-2019
respectively. The smooth lines and shaded areas represent the Kernel-regression at 95% significance
level. The thin lines show the monthly time series of GEM after removing annual cycles with
amplitudes of 0.49 ng m$^{-3}$, 0.23 ngm$^{-3}$, 0.17 ng m$^{-3}$, 0.30 ng m$^{-3}$, 22 ng m$^{-3}$, and 0.25 ng m$^{-3}$ respectively
for Amderma, Andøya, Mace Head, Villum, Waldhof, and Zeppelin. The annual cycle was calculated
based on seasonality of the time series decomposition. *An individual plot regression for each station
is presented in Figure 1S.






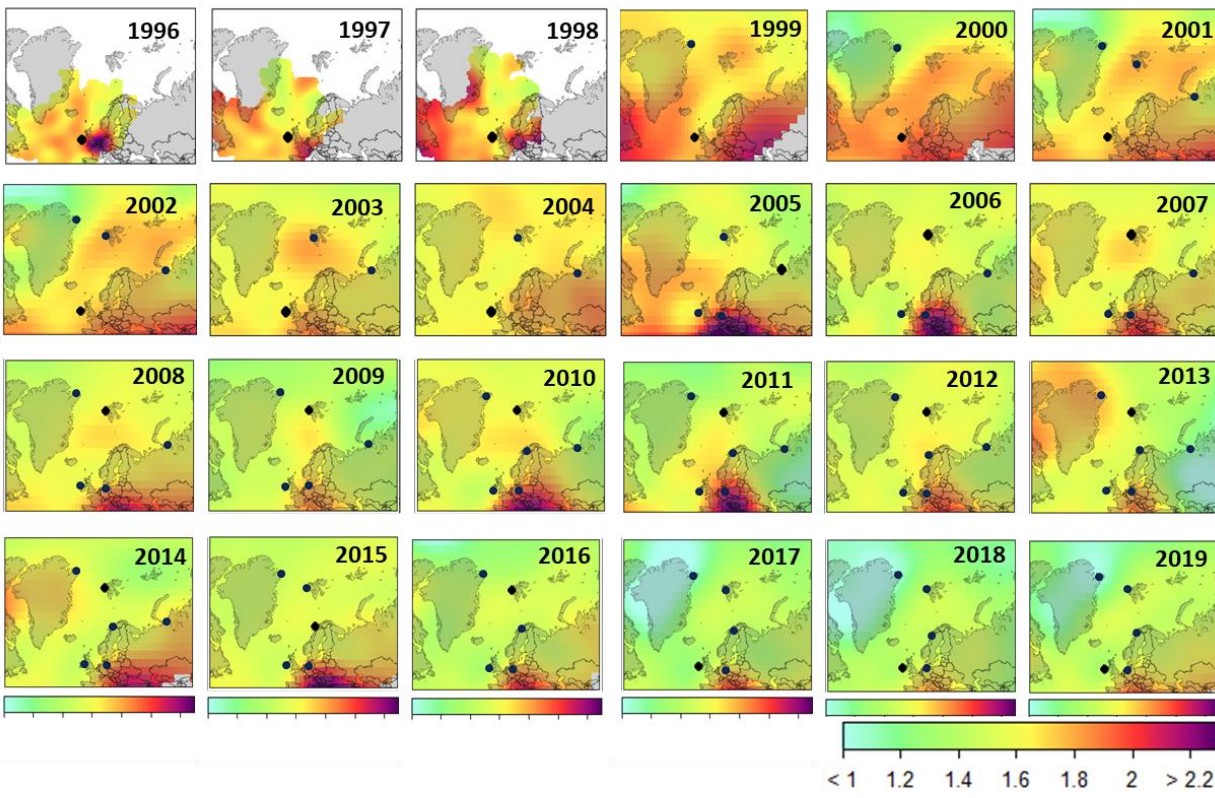


Figure 4: Concentration level (concentration-weighted trajectory) of GEM (ng m$^{-3}$) based on the mercury concentration associated to its reanalysis backward trajectory at Amderma, Andøya, Mace Head, Villum, Waldhof, and Zeppelin. *The black dots show the arriving point (stations) considered for each year.









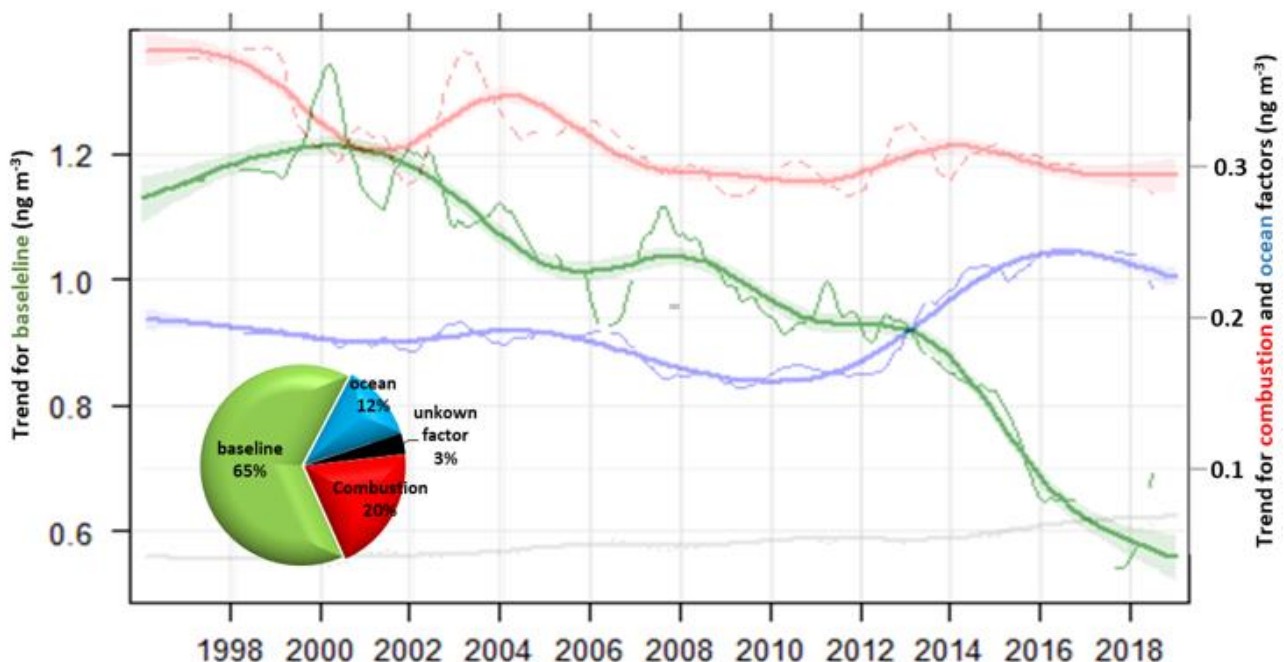


Figure 5: Time series (thin lines) and percentile average contribution (pie) of factors solved by PMF

in the GEM reconstruction for Mace Head from 1996 to 2019, baseline (green) combustion (red),

ocean (blue) and unknown factor (grey). The smooth lines and shaded areas represent the Kernel-

regression at 95% significance level. The thin lines show the monthly time series with annual cycles

removed.

594

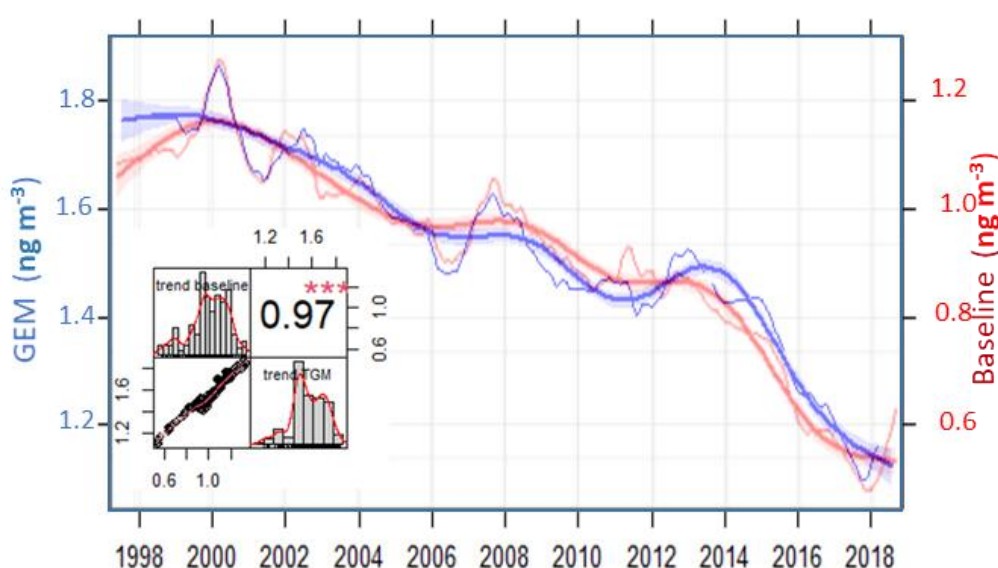

Figure 6: Downward trend of GEM (blue) and baseline factor (red) at Mace Head. The smooth lines and shaded areas represent the Kernel-regression and 95% significance level. The thin lines show the monthly time series with annual cycles removed. On the bottom right it is presented the correlation regression with the distribution of each variable and the value of the correlation plus the significance level as stars. p-values (0.001) <=> symbols("***").

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
