# Peer review of "poral trends of 25 years of measurements. 3 4 Danilo Custódio1\*, Franz Slemr2, Katrine Aspmo Pfaffhuber3, T. Gerard Spain4, Fidel F. Pankratov 5, Iana 5 Strigunova6, a, Koketso Molepo1, Henrik Skov7, Joha"

_Atmospheric Chemistry and Physics, 2021_

## Referee Comment (RC2)

Review ACP Custodio et al

Overall, it is not clear to me the purpose of this paper and how it is providing new information. The datasets presented are rich and have the potential to provide interesting new results, but as of now there is little in the paper that is new. A lot of previous work related to what is presented is not discussed. I elaborate on the main issues first and then present line-by-line comments.

**Major and General Comments**

**Discussion of the Tekran instrumentation and uncertainty should be improved, related to whether TGM or GEM is being measured.**
Line 134: "usually" – which of these stations' instruments have this filter operating and which do not?

Line 137: "probably" – what does this mean quantitatively? Slemr et al. (2016) describes the CARIBIC aircraft measurements in the upper troposphere/lower stratosphere which are specifically in a low-humidity but high ozone environment. How do the conclusions regarding GOM compound collection from Slemr et al. (2016) apply to the surface stations, especially coastal ones, in this study?

Line 139-140: "soda-lime filter.. is known to capture GOM" – please add citations here. Gustin et al. (2021) state: "The Global Mercury Observation System standard operating procedure states that a soda lime trap in front of the Tekran 2537 removes GOM, though this has not been adequately tested."

In particular, it is not actually known whether >95% of TGM is GEM outside of polar depletion events (the discussion at the end of the sampling sites part of Section 2, lines 143-146). Please add a reference to at least one of the papers and reviews discussing the major uncertainties measuring gaseous oxidized mercury (e.g., Marusczak et al., 2017; Lyman et al., 2020; Gustin et al., 2015, etc.) as it relates to uncertainties in the operationally-defined TGM Tekran measurements. I find that Lyman et al. 2020 summarize it better than I could myself and quote from them here: "it remains unclear whether Tekran 2537 and similar mercury analyzers without upstream processing equipment measure total gas-phase mercury or only elemental mercury. Thus, these measurements are also operationally defined. We use the acronym GEM (gas-phase elemental mercury) to describe them, though they likely include GEM and some portion of $Hg^{II}$ compounds that exist in the atmosphere, and the amount of $Hg^{II}$ they include likely depends on the sampling configuration and the chemical and physical conditions of the atmosphere. Also, when KCl denuders are used upstream of elemental mercury analyzers, some of the captured $Hg^{II}$ is reduced to elemental mercury and measured in that form (Lyman et al., 2010). Some have used the acronym TGM (total gaseous mercury), rather than GEM, but, to our understanding, no information exists about the percentage of gas-phase $Hg^{II}$ that is analyzed by elemental mercury analyzers with an upsteam KCl denuder or without upstream sample processing. Many have asserted that this issue is inconsequential, because atmospheric $Hg^{II}$ concentrations are low

relative to Hg$^0$ (Ci et al., 2011; Fu et al., 2012a), but this assertion has been shown to be inaccurate in some environments (Fu et al., 2015; Obrist et al., 2011; Swartzendruber et al., 2006; Weiss-Penzias et al., 2009), especially when the low bias in KCl-denuder based GOM measurements is taken into account (Huang et al., 2013; Lyman et al., 2016)."

Overall, I am confused when the methods state that it is concluded only GEM is being measured but then all data is treated as TGM. Should it be GEM and not TGM? Interpreting the observations as GEM is consistent with other studies (e.g., Travnikov et al., 2017) but a more thorough discussion of the caveats is warranted.

**Clarify and provide more information on the methods used.** While I understand not wanting to repeat too much from previous work, as the sub-sections "Back-Trajectory Analysis" and "Concentration-weighted trajectories (CWT) and probability mass function (PMF) models" are written now the reader cannot understand what is being done. More information must be provided so that what is in the current paper becomes sufficient, and only the reader who wants to learn more specific details needs to look to Custodio et al. (2020).

Back Trajectory analysis: I found this section very confusing. Perhaps a schematic (potentially in the supporting information) could be added showing each step of this method. Please provide more information on when the trajectories are initiated. Is it every 12 hours for all days and years of data or a single year of data? If it is a single year, which specific year is it? Please explain and justify all choices made for the HYSPLIT simulations carefully including the arrival times of midnight and 12pm (is this local time or UTC?) and arrival heights of 50 and 500m. Combining the concentration-weighted trajectories for all 5 sites, rather than analyzing them on a site-by-site basis, seems like it waters down the results and adds uncertainty especially as the number of sites with data is different for different years.

Concentration-weighted trajectories are described in the "Back-Trajectory Analysis" section, yet they are in the title of the following section.

Line 169: I am much more familiar with PMF standing for the EPA positive matrix factorization method. In Custodio et al. (2020), PMF is used as an abbreviation for positive matrix factorization, not "probability mass function".

CWT and PMF section: these two paragraphs are quite confusing with some very long sentences that are difficult to follow.

The methods in Section 3.2 are also in need of additional explanation, including "Kernel-regression", "LSQF" (including defining this acronym in the main text), how the annual cycle was removed from Figure 3, and how confidence intervals were calculated (or what the trend ± values actually mean: are the ± values the standard deviation, standard error, 95% confidence interval from bootstrapping?). It is not clear from the description here or in Figure 3 that the regression is being performed on monthly medians. The standard deviation or confidence intervals reported for trends in this paper are much narrower than in previous studies cited (e.g., Cole et al., 2013). Why is this, and how do the methods in calculating the confidence intervals in Cole et al. (2013) differ from this study?

**Improve statistical analysis of the observations and distinguish what is new in this paper, including relative to Custodio et al. (2020).**

In the opening section of the results (Lines 194 – 215), please provide quantitative metrics of the skewness and kurtosis of the distributions, and differences between sites or average vs. median rather than qualitative descriptions. Can you quantitatively define pollution events, such that it is clear that "Events with local and regional pollution are also missing at Andøya"? I also am confused about how Weigelt et al. (2013) is a reference for "frequent events with local and regional pollution" at Waldhof. As far as I can tell they mention a single time that a local combustion plume was observed and overall put Waldhof on par with other rural background sites. Again, a quantitative definition of pollution events would be helpful here.

The 3.1 seasonal variation section does not seem to add any new information as seasonal patterns at these sites have been published in other studies previously. What would be new is if you could add more detailed and rigorous comparison of the seasonality between sites (they are "similar") or from the longer time record if the seasonal cycle has been changing over time at any individual site. An example of a more detailed analysis of observed seasonality at Arctic sites including statistical tests of the significance of differences between seasons and between sites' seasonality is presented in Angot et al. (2016). The amplitude of the annual cycle at each site is only referenced in the caption of Figure 3. This should be in the main text along with an explanation of its calculation. In section 3.2, the comparison of the trend at each site to Mace Head because it is the "baseline trend" is not explained. Table 1 is confusing as it currently stands, please add more explanation in the text and in its caption. Please also explain why and how the different time periods of trends are calculated for Mace Head. Similarly in this section, the comparison of trends between sites are described qualitatively (e.g., "similar", "somewhat higher", "substantially higher") rather than quantitatively.
The paper would be stronger if the PMF analysis was performed at other sites besides Mace Head, where it has been done before for a subset of the time period presented.

**The conclusions drawn in the paper do not seem to be supported by the analysis.**
The discussion in Lines 281-288 is speculative. I do not have enough information to evaluate this argument.
- While the back-trajectory analysis may show transport from Greenland, how do we know from observations that the atmospheric mercury there was elevated?
- I would need to see a timeseries of the AO and NAO plotted against the mercury concentrations in order to evaluate how extreme the AO was for those 2 particular years and its relationship to mercury concentrations over the full time period.
- Hawkings et al. (2021) is reporting measurements from 2018 as far as I can tell. How is this related to 2010 and 2013?

The discussion in lines 316 – 334 referencing specific policy actions and their relationship to observed trends is lacking evidence. Some specifics:
1) Minamata convention signing in 2013: The assumption that policies were already implemented at this point before the treaty went into force, and that this would be observable at Mace Head, ignores previous work done showing the difficulty in linking observations to the Minamata Convention and other policies (e.g., Selin, 2014; Giang et al., 2018). Whether emissions may have already changed at the timing of the treaty

signing could be investigated directly by examining emissions inventories over this time period (e.g., Streets et al., 2019 global inventory for 2010-2015; GMA2018 global inventory for year 2015 (AMAP/UNEP 2019); Leclerc et al., 2019 – 2000 – 2014 for the EU;)

2) air pollution policy in the United States: I assume this is referencing what eventually became the MATS rule (https://www.epa.gov/mats) which was tied up in the courts for a number of years (and potentially still is as far as I can tell). Zhang et al. (2016), while the inventory itself is only presented until 2010, did analyze changes in Hg emissions from US coal-fired power plants from 2005 to 2015 (see Figure 1) and attributed a significant portion of the decline to mercury-specific control technologies resulting from the planned MATS rule coming into force. However, after 2016, when the MATS rule became tied up in the courts, I would not have expected coal-fired utilities to continue operating expensive mercury-specific controls if they did not need to. It would be possible to investigate this by applying similar techniques of Zhang et al. (2016) to later years of data for US power plants.

3) The 2007 policy banning mercury in non-electrical measuring devices in the EU could be quantified based on existing inventories (e.g., Horowitz et al., 2014; Streets et al., 2017, 2019; GMA from AMAP/UNEP 2013 or AMAP/UNEP 2019 reports) rather than speculated. From these studies, this sector is not a large portion of the mercury emissions and thus I doubt it would be able to be observable in atmospheric trends.

The paper discusses downward trends and states them as such in the conclusions in lines 426 – 427, but at four out of the six sites the trends shown in percentage form are not statistically significant (confidence interval includes 0% yr$^{-1}$). In Table 1 the trends are all statistically significant – how is this the case? As I mentioned earlier, the standard deviations(?) or confidence intervals of the trends shown in Table 1 are much narrower than in previous studies like Cole et al. (2013). Much past work has discussed how Arctic sites do not have a decreasing trend unlike mid-latitude sites. Skov et al. (2017) performed a more detailed statistical analysis at Villum from 1999 – 2017, finding no significant annual trend in GEM, and their comparison with a model allowed for more convincing attribution of emissions to the mercury concentrations observed.

The "seemingly non-monotonic downward trends with inter-annual ups and downs are not well-explained" also ignores previously published work. The attribution of trends to anthropogenic emissions is not convincing. Other potential mechanisms are not discussed and natural sources and sinks are referenced only briefly. A detailed review of observed trends and potential mechanisms including anthropogenic emissions, natural emissions/re-emissions, sinks, and oxidative capacity is also provided in Lyman et al. (2020) and references therein. Declines in anthropogenic emissions are referencing studies who only estimated anthropogenic emissions until 2008 (Streets et al., 2011) or 2010 (Horowitz et al., 2014; Zhang et al., 2016). There is no reference to a more recent study (Streets et al., 2019) that estimated mercury emissions as well as modeled their impact on concentrations from 2010 to 2015, including discussions on a regional basis. Another recent study that is not referenced here (Wang and Mao, 2021) specifically examined how trends in emissions from the EDGARv4.tox2 inventory

(1970 through year 2012; Muntean et al., 2018) and the location of Arctic fronts interacted to affect transport of emissions to the Arctic. Finally, interannual variability in Arctic atmospheric mercury from 1979 to 2008 was also investigated in Fisher et al. (2012) and was related to specific environmental and climatic factors.

**Figures are not adequately described prior to when interpretations based off them are given.**
- Figure 2: interpretation is given in Lines 218 – 220, but it is not until Line 237-238 that it is explained what is specifically in the Figure. This explanation is also not linked to the methods section where more detail is given on the density map. However, there is still not enough information – for example, what is the size of the grid cells in Figure 2? how was this chosen?
- Figure 4: Interpretations of Figure 4 are given in Lines 282, 297-299, 302-306, and 312-313, but the figure is introduced / described in Lines 352-353. Please move the description of Figure 4 from Lines 352-353 before any interpretations of the Figure are given.

**Line-by-line comments**

line 31: is there only a single probabilistic source contribution factor?

Line 35: Is this superscript 1 a citation or footnote? Please clarify whether these are results you have found or you are citing other esults.

Line 38: I would appreciate if the "other atmospheric trace gases" are listed here or at least a few examples are given.

Lines 37-39: This sentence is confusingly worded. Please clarify and potentially split into multiple shorter sentences.

Line 40: Instead of "accessed" I believe you mean to say "assessed"

Line 42, Line 68: I similarly would like to know what the long-lived anthropogenic species are, or at least a few examples.

Line 71: In this discussion of regional and hemispheric trends, the mention of global trends doesn't seem to follow. Is there a reference that hypothesizes or inventory that estimates that global anthropogenic mercury emissions are decreasing? Perhaps this should be replaced by "hemispheric" instead.

Lines 72-73: "still not unknown sources" is unclear. I thought it would make more sense to say "still unknown sources", but how could one do a time series analysis of unknown sources? Overall this sentence is confusing.

Line 80: "On this raw" – I do not know what this means.

Line 81: "baseline factor" – the meaning of this is not explained. Perhaps this should not be mentioned in the introduction, but explained instead in the results and also methods. Otherwise, please explain what this means in a way that does not require reading the methods first.

Line 84-85: I would add "observed" in front of "atmospheric mercury temporal variability" to distinguish it from models that you are trying to constrain. This sentence is also a bit vague – what kind of temporal variability (seasonal? interannual? or other analysis) and what kind of cycling predictions by the model? (e.g., concentrations, deposition?). I have not again seen a reference to constraining models outside of the introduction.

Section 2 – Sampling Sites: A map with the stations labeled would be helpful. Although the latitude and longitude are given in individual paragraphs, having this information in a table or on the map would be helpful so it is all in one place. Sometimes the altitude of the stations is not given. Similarly the altitudes could be included in a table or map label.

Line 96-97: At this point, we do not have the latitudes of the stations and it would be helpful to have seen these (or the locations on a map) for the comparison of Andøya and Amderma. Is it known from previous analyses or your own analysis that the mercury concentrations at Andøya are more similar to those observed at other mid-latitude sites? (Angot et al., 2016 found this, for example, in their analysis of seasonality of Arctic sites). It seems your own analysis from the seasonality in your Figure 2 also supports this, but at this point in the paper the classification of Arctic vs. Midlatitude sites is not well-motivated. Explain why this classification will be useful.

Line 109, 115, 120: Altitudes are not given for the Waldhof, Villum, or Andøya sites.

Line 120-122: To me, it does not add anything to mention ALOMAR here.

Line 130: At the end of every paragraph introducing the measurement sites, I was wondering what instrument they were operating. It would be helpful to give the reader a heads up (perhaps at the end of the first paragraph) that all sites operated Tekran instruments and more detail will be given at the end of the section.

Line 152: write out the acronym for HYSPLIT the first time it is used.

153: make clear that Stein et al. (2015) is the reference for HYSPLIT model. As it is written now it sounds like Stein et al. (2015) describes the method of choosing the trajectory heights and time initiations, but these are choices made by the authors here which must be explained.

Line 171-173: this sentence should be rewritten and split into multiple sentences for clarity. Right now it seems like some words are missing for it to make sense.

Lines 179-181: this is also a very long and hard to understand sentence that should be split up for clarity.

Lines 188 – 192: rather than list the number of days of data separately from the station names, include this information in a table. The number of months are listed in Table 1. What information does the number of days add? is the data averaged daily? Are daily averages what is summarized in Figure 1?

Figure 1. please state in the caption the temporal frequency of the measurements presented (e.g., daily averages?).

Line 195 – 199: This sentence is also very long and should be split into multiple sentences for clarity. Also, what is the detection limit of the Tekran instruments at these sites? Are minima of 0.0 vs. 0.1 ng m$^{-3}$ real and different? It would be more useful to know the number of depletion events in each year rather than just the descriptor "frequently". Referencing studies of depletion events in and around these particular sites should happen sooner in this paragraph. A detailed analysis of changes in the frequency and timing of depletion events as well as comparison with previous studies is presented in Angot et al. (2016), referencing in addition to the studies already cited here, Berg et al. (2013) and Chen et al. (2015).

Line 202 – 203: I am confused how the altitude of the Zeppelin station is relevant here as the area around the station and higher-altitude areas near it can also be snow-covered.

Line 261: What does "regional emissions" mean? Is this anthropogenic emissions? Natural/legacy re-emissions from land?

Line 279-280: I do not understand the meaning of this sentence and how it relates to the previous sentence. Changing the discussion from qualitative to quantitative might help.

Line 289-291: Please add a citation for this sentence.

Line 298-299: Please show and cite specific anthropogenic emission estimates for this region, otherwise this sounds like speculation.

Line 300: Bringing up uncertainties in the HYSPLIT modeling approach is critical and there is not enough explanation of what you mean here. I do not understand the shift to the South that is being described.

Line 301-302: Please add a specific citation showing the "geolocation of TGM sources in Europe." A figure showing the locations of major point sources or gridded anthropogenic Hg emissions inventory for the region would be helpful.

Line 303-304: The explanation given in Lines 281-288 seems to conflict with what is shown and described in Figure 4 here. The earlier explanation seems to be specific to 2013 and potentially

2010-2014, but similarly high levels are shown in Figure 4 over Canada and Greenland for 1997-2000 and 2005. Please clarify.

Line 307: This is the only time sinks are mentioned as affecting the concentration of mercury in the atmosphere. This should be expanded in more detail.

Line 311: "This phenomenon can be explained only by reductions in global atmospheric mercury sources" : I am not yet convinced. Evidence must be presented to back up this claim. More discussion of specific anthropogenic emissions inventories and estimates of emissions from natural sources, legacy re-emissions, as well as sinks, chemistry, etc. is needed.

Lines 310-311 and 312-313: A quantitative assessment of the trends in the concentration-weighted trajectory would be more helpful than qualitatively referring to the colors on the figure.

Lines 314-315: The trend at Villum seems very different from the that at Zeppelin. What are they affected by if not anthropogenic emissions from continental Europe? Please reference here prior work that has been done in this space (e.g., Fisher et al., 2012; Skov et al., 2017; Angot et al., 2016; Wang and Mao, 2021).

Line 321-322: Landfills are not a significant source of mercury. Horowitz et al. (2014) reviewed measurements of emissions of mercury from landfills and found they can be treated as a long-term sink of mercury on centuries-long timescales. as their emissions of mercury are so small the lifetime is on the order of 20,000 years.

Line 341-342: This may be true, but there have been recent studies on other natural processes, e.g., Hg uptake to land (Zhou et al., 2021; Jiskra et al., 2018) driving seasonality and trends.

Lines 348-350: Where is this shown?

Lines 357-358: How do you know that the North Atlantic a sink of anthropogenic pollutants? The North Atlantic is a hot spot of net evasion in Soerensen et al. (2010) and Zhang et al. (2019).

Line 359-361: This hypothesis seems like it would be testable by performing chemical transport modeling experiments.

Lines 364-365: I would like to see analysis separating out these two processes, transport and the concentration in transported air. This might be similar to what was done in Wang and Mao (2021).

Line 368: The discussion of factors is a surprise. Please introduce the PMF analysis here before going into the factors. It might need to be a separate section.

Lines 369-373: I don't understand how the trends in a factor from 2013 – 2018 can be explained by declines in emissions that were estimated only until year 2008 (Streets et al., 2011) and year 2010 (Zhang et al., 2016). Please explain.

Line 376: there is not enough information on the baseline factor given.

Line 377: What anthropogenic species?

Line 379: This is an extremely high correlation coefficient. I need to know more about what went into this factor and what we can learn from it if it is explaining 94% ($r^2$) of the variance in observed concentrations at Mace Head.

Line 380-381: More explanation is needed as to how Figure 4 has to do with the anthropogenic species factor.

Lines 383 – 397: I am very confused by this section. Explaining the baseline factor in detail would help with that.

Line 399-400: A different reference for the residence time of mercury in the ocean would be more appropriate (e.g., Amos et al., 2014?)

Lines 400-402: Additional references about the oceanic mercury influence on atmospheric mercury due to anthropogenic activity should be cited here like Soerensen et al. (2012); Sunderland and Mason, 2007; Sonke et al. (2018); Cossa et al. (2018); etc.

Lines 403-305: I don't think this adds much to the paper, unless you add references to studies that have examined some of these processes for mercury specifically.

Lines 410-411: This conundrum has been explained by other studies. The analysis presented here does not include any estimates of anthropogenic emissions to corroborate that the current study also shows this.

Line 426-427: I think there is a typo, is the trend actually 4 ±16 % per year or should it be 4 ±1.6% per year? Also, typo Waldholf -> Waldhof

Line 421-422: This is not appropriately caveated given other studies on how difficult it is to observe the impact of the Minamata Convention and other policies (e.g., Selin, 2014; Giang et al., 2018).

Additional references cited in this review, not already cited by Custodio et al. (2021):
Gustin et al. (2021): https://www.mdpi.com/2073-4433/12/1/73

Lyman et al. 2020:
https://www.sciencedirect.com/science/article/pii/S0048969719355706#bb1685

Marussczak et al. 2017: https://pubs.acs.org/doi/10.1021/acs.est.6b04999

Gustin et al. 2015: https://acp.copernicus.org/articles/15/5697/2015/

Travnikov et al., 2017: https://acp.copernicus.org/articles/17/5271/2017/

Angot et al. (2016): https://acp.copernicus.org/articles/16/10735/2016/

Selin (2014): https://setac.onlinelibrary.wiley.com/doi/full/10.1002/etc.2374

Giang et al. (2018): https://pubs.rsc.org/en/content/articlehtml/2018/em/c8em00268a

Streets et al. (2019): https://www.sciencedirect.com/science/article/pii/S1352231018308884?via%3Dihub

Leclerc et al. (2019): https://www.sciencedirect.com/science/article/pii/S0160412018330101

AMAP./UNEP 2013: https://www.amap.no/documents/download/1265/inline

Streets et al. (2017): https://pubs.acs.org/doi/abs/10.1021/acs.est.7b00451

Wang and Mao (2021): https://www.sciencedirect.com/science/article/pii/S1352231020307603?via%3Dihub

Muntean et al. (2018): https://www.sciencedirect.com/science/article/pii/S1352231018302425

Fisher et al. (2012): https://www.nature.com/articles/ngeo1478

Berg et al. (2013): https://acp.copernicus.org/articles/13/6575/2013/

Chen et al. (2015) https://agupubs.onlinelibrary.wiley.com/doi/full/10.1002/2015GL064051

Soerensen et al. (2010): https://pubs.acs.org/doi/10.1021/es102032g

Zhang et al (2019): https://pubs.acs.org/doi/10.1021/acs.est.8b06205

Amos et al. (2014): https://pubs.acs.org/doi/abs/10.1021/es502134t

Soerensen et al. (2012): https://agupubs.onlinelibrary.wiley.com/doi/full/10.1029/2012GL053736

Sunderland and Mason, 2007: https://agupubs.onlinelibrary.wiley.com/doi/full/10.1029/2006GB002876

Sonke et al. (2018): https://www.pnas.org/content/115/50/E11586

Cossa et al. (2018): https://bg.copernicus.org/articles/15/2309/2018/

Zhou et al. (2021): https://www.nature.com/articles/s43017-021-00146-y?proof=t+target%3D

Jiskra et al. (2018): https://www.nature.com/articles/s41561-018-0078-8

---

## Author Comment (AC1)

Response to Anonymous Referee #1
* * *
Referee comment on "Odds and ends of atmospheric mercury in Europe and over northern Atlantic Ocean: Temporal trends of 25 years of measurements" by Danilo Custódio et al., Atmos. Chem. Phys. Discuss., https://doi.org/10.5194/acp-2021-753-RC1, 2021
* * *
This paper presents trends in total gaseous mercury concentrations in Europe and the North Atlantic Ocean and the regional sources affecting TGM inferred from concentrationweighted trajectory (CWT) analysis and the Positive Matrix Factorization (PMF) model. Ten to twenty-five years of TGM data at six locations were analyzed in the study. Given the long term data available, there needs to be a more detailed and deeper analysis of the data than the one currently presented. The paper summarized the distribution and general statistics for the 10-25 year period. There should be more detailed analysis comparing TGM data over various time periods and among the sites and examining changes in the frequency of Hg depletion events, etc. as well as the explanations for the TGM variability. Long term declines in TGM appear to be related to decreasing anthropogenic Hg emissions; however, Hg emissions data from the region were not presented in the paper.

Response: The authors thank the comments. New statement was added in order to improve the article and a more detailed analysis was included.

One of the major point highlighted by the reviewer is the necessity of discussing anthropogenic emission. The author agree on the relevance of such discussion. An accurate emissions inventory is essential for interpreting trends in atmospheric concentrations and assessing the effectiveness of mercury pollution control policies. However, the observed GEM/TGM trend is not consistent with the global anthropogenic emissions inventory. As reported by Lyman et al. (2020 and references in) the uncertainties in inventories range from -33% to 60%.

The emission issue was intensively discussed during the article preparation among the authors. Indeed, the is a conundrum in the trend displayed by the emission inventory.

Listen different partners the conclusion was that having trend analysis for the emission inventory could be not suitable for comparison with the observed one. Due to gaps, errors and inconsistencies in the data from different periods, I would be afraid that a trend evaluation of atmospheric mercury based on inventories could be misleading. Emission inventory relies on data provided by countries and emission sectors.

There is a protocol, however it explicitly informs that national agencies can use its on method.

Many countries just do not provide data, so it is estimated. It used different parameters to estimate, like population density, vegetation, ecosystem. Sometimes there is even interpolation when data are not available for some sectors in some regions or period.

Even in Europe, the emission assessment for sectors as power plants, industrial installation, oil and gas production sites, waste units can follow proxies and methods that have changed over time. (Thus, the sector classification can be changed).

Strong deviations can be observed for following years as for example, 2015-2016 when the spatial resolution of 1/10° x 1/20° in longitude and latitude was applied to 30° W – 60° E and 30° N – 72°N in many atmospheric pollutant species inventories.

In addition, even emission estimated based on remote sensing techniques have be tremendously improved.

Since the emission inventory are more committed with a better description and accuracy of sources rather than its method coherence throw the years, and the gathering of data has been constantly improved, the apparent trend displayed by emissions inventory can be

mainly caused by the improvement in the emission catalogue. This is the main argument for not having the emission data discussed in the manuscript. It may not represent the observed trend.

Others comments above mentioned by the reviewer are replied in the specific comments.

The detailed methodology for the CWT and PMF analysis are missing. It is unclear why back trajectory data are combined for the five sites rather than analyzing the data for the sites individually. CWT results likely differ from one site to another. The PMF analysis in this paper was very similar to the one conducted by Custodio et al. (2020) published in this special issue for the 5-year period at Mace Head, and does not seem to provide substantially new results and insights.

Response: The source apportionment and CWT are only methods exploited in order to bring insights into the trend evaluation, which is the aim of the article. Rather than a detailed methodology description, which will take an extensively part of the article, the author stands for addressing the references of the full description of the method. However, the authors agree that further description of the method helps the reader understanding and evaluate the finding. Therefore, we appreciate the comment and new information in the factorization and factors obtained were added.

The PMF results are also not as detailed as those in Custodio et al. (2020) despite a longer time-series in this study.

Response: A longer times series has no analytical or computation implication for the initial condition considered in the PMF factorization (in terms of factors, seeds, analytical uncertainties). It basically increases the number of equations, propagating the factors backwards in time. However, key failures as residual and reconstruction performance are important for the reader to evaluate the mass reconstruction. This information was added in the new version.

The baseline factor extracted from the PMF model needs to be described in detail given that this factor explains the largest proportion of the TGM variability. It was not clear from the paper which pollutant markers were used to assign a PMF factor to the baseline factor. The lack of a clear definition of the baseline factor makes it challenging to evaluate the effectiveness of mercury control measures that have been implemented. In my view, the source apportionment results should provide a better understanding of the different anthropogenic source sectors contributing to TGM and whether their contributions have changed over time, and if there are emerging Hg sources that we need to be concerned about. The role of re-emissions of previously deposited Hg is also not well understood.

Response: Baseline factor refers to a background factor, a factor where TGM display an established well mixing ratio. The authors are taking the reviewer comment into account and improving the factor description.

The new version of the manuscript brings the full profile of the sources obtained from the factorization, which reconstructed the TGM mass. The chemical profile obtained for each factor is now presented on the supplement.

The authors understand the importance of the re-emission role; however, it is important to highlight that we are dealing with a receptor model. The solution is constrained by the data that you have, which will allow the propagation of the eigenvector. The axis rotations without a re-emission marker will not solver such "source". The same can be said about better speciation of sources. We are totally in line with the reviewer idea that a better understanding of the different anthropogenic source sectors contributing to TGM is needed and should be sought. However, more source marker is needed to develop and permit better source apportionment. Custodio et al. (2020) give a contribution on decluttering atmospheric mercury sources by PMF, but their solution is far from being the ultimate. In addition, the authors want to declare that there is no improvement in the source apportionment method used in this study. We only applied the method presented by

Custodio et al. (2020) to exploit the TGM trend. This study was conceptualised as Part II of the aforementioned article, even been the authors not being the same.

Specific comments

Abstract. It is not necessary to describe the PMF model in the abstract. It is more meaningful to focus on the PMF factors extracted and what sources they represent. Please also clarify the last sentence of the abstract.

*Response: Thanks for the comments. The abstract was reworded.*

Lines 57-65. The updated review paper on atmospheric mercury (Lyman et al., 2020) should be discussed and cited. This review paper summarized trends in TGM and provided potential explanations for the trends.

Response: We appreciated the comment. The aforementioned reference was considered in the new version.

Lines 67-70. The paper by Custodio et al. (2020) is a five year study of the TGM during the 2013-2018 period, while historical mercury releases are discussed in Horowitz et al. (2014). The timescales are very different; thus it does not seem reasonable to attribute the TGM decrease over that period to historical changes in mercury releases.

*Response: The aforementioned statement stands for compliance with those articles findings rather than a trend comparison.*

Lines 71-73. There have been several studies examining long-term trends in TGM and relating that to mercury emissions trends. Please discuss and reference these studies and provide some explanations on how this study is different.

Response: As far the authors know, this is the first publication deploying time series decomposition, combining Lagrangian transport and receptor model in the evaluation of TGM/GEM trend.

Lines 77-79. Were 5-day trajectories analyzed over the 25-year period? Please clarify.

Response: We analysed trajectories arriving at Mace Head with a backward length of 120 h. Two trajectories per day for a period of 25 years. An improved statement was added in the section 2 subsection, "back-trajectory".

Line 80. Delete "On this raw,"

Response: Sentence reworded. Thanks.

Line 81. Please clarify the meaning of "baseline factor"

Response: A factor with a high load of long-lived anthropogenic species was labelled by Custodio et al. (2020) as a baseline factor. Such a name was picked up because it was the main factor at Mace Head station, a background site with a lower impact of nearby anthropogenic sources. (The factor was labelled as "Aged TGM", however, it was renamed baseline factor as requested by the reviewer community in a previous paper). Further consideration on the factor meaning and interpretation is given in lines 377-384.

Line 128. It is useful to label the TGM sites on a map and show the spatial distribution of the mercury point sources in Europe.

Response: Thanks for the comment. The sites' names were added in Figure 2.

Line 130-146. Please clarify which analyzer model was used to measure TGM and the sampling interval of the TGM data. There should also be some discussion on quality control, calibration and maintenance of the analyzers.

Response: All sites used Tekran 2537 (A and/or B) instrument. A new statement is presented.

Line 150. Was the CWT analysis conducted at five or six stations? Long term TGM data are available at six stations as mentioned in the previous section.

Response: The CWT was performed at six stations. The mistake was corrected. Thanks.

Lines 151-152. Some justification for the back trajectory model parameters are needed. It seems that two trajectories per day is not sufficient given that TGM data are available every 5 minutes. Why did you choose a start time of 0:00 and 12:00 local time and two arrival heights?

Response: For the CWT, it was considered TGM/GEM in a time resolution of 12h. As well, it was obtained trajectories arriving at the stations representing the average time of 12h. A higher resolution would require intense computing power since the time series is quite long and has six combined sites. In addition, considering the trajectory uncertainties, it was not observed significant difference among trajectories closely collocated in time. Thanks for the comment. The statement was reworded.

Line 161. "For a 120-hour trajectory duration, 84 trajectory segment end points were calculated." It's not clear how the 84 end points were obtained. There should be 120 trajectory segment endpoints for each 120 h trajectory.

Response: 84 is the maximum in segments that hysplit can provide (1.42h/segment).

Line 169. PMF stands for Positive Matrix Factorization.

Response: Indeed. Thanks.

Lines 171-175. It is unclear as to which year(s) of data the PMF model was applied to. Why was PMF applied to only the Mace Head site and not the other five sites? If the purpose is to examine changes in the sources affecting the TGM sites, then PMF should be applied to the yearly data or different time periods rather than the 25-year period.

Response: PMF is applied to daily data. We reconstructed daily data at Mace Head for a period of 25 years. It was performed only for Mace Head because the site has the best database, disposing of a better tracer gases speciation (markers).

Lines 177-184. The introduction to the CWT method should have been discussed in the back trajectory section where the CWT calculation was described.

Response: The sections were reworded. Thanks.

Figure 1. The time-series shown is not very meaningful as it is difficult to see the variations due to the large number of data points.

Response: Figure 1 is a summary dedicated to presenting data distribution, central tendency, and timely availability of data. Since the dataset is trended, any comparison should consider the period of missing data. The period with a lack of data is displayed in the aforementioned figure. In addition, the time variability as seasonality and trend (which need further data decluttering) are discussed and showed in later figures.

Lines 188-215. A more detailed analysis of the TGM data is needed given that there are almost 25 years of data measured at six sites. For example, a comparison of the TGM data over the three time periods, comparisons between sites, changes in the frequency of Hg depletion events, etc.

Response: *The presented analysis of the measurements compares the sites only as far as needed for the application of the CWT and PMF modeling. A detailed analysis proposed by the reviewer would need a separate paper.*

In addition Hg depletion is taken in this study as a modulated phenomenon. Indeed, changes in the frequency of depletion phenomena and in the strength of its events would be an essential component in the atmospheric mercury reducibility and should be decluttering in future investigations. Comparison and evaluation of the non-monotonic trend in different periods is presented in section 3.2

Line 218. Please label the site names on one of the maps.

Response: Label with the name of sites added in figure 2.

Line 252. LSQF has not been defined.

Response: Indeed. Thanks.

Lines 264-273. Some studies in North America (Weiss-Penzias et al., 2016) and Mace Head (Weigelt et al., 2015) observed a smaller decreasing trend in TGM during the recent decade. Was this observed at the six sites analyzed in this study? Explanations on long term TGM trends were provided by Lyman et al. (2020) and should be discussed in this paper.

Response: Yes, the period studied by Weiss-Penzias et al. and Weigelt et al. 2015 is contemplated in this study and new insights are stated. Lyman et al. (2020) is cited and discussed in section 3.2 of the new version.

Lines 281-294. The discussion of the trend at Villum is quite interesting. From Figure 3, there was a substantial increase in TGM from 2010 to 2014, which you attribute to melting of sea ice and subsequent GEM emissions. However, what caused the abrupt decline in TGM from 1.7 to 1.0 ng/m3 after 2014? Was that related to changes in sea ice?

Response: The abrupt decline corresponds mainly to the re-establishment of atmospheric circulation normality, and consequently, stopping the ice mass discharge. However, I am afraid that linking the abrupt decline in mercury at Villum after 2014 to sea ice would sound a bit speculative in the face of the presented results. More data would need to support such a conclusion.

Line 295. Did you analyze the CWT results for each site separately? The probable source regions are likely different depending on the site. Why were the CWT results combined for all sites?

Response: Combining back trajectories from different sites enhance and allows a broader evaluation of hotspots and regional sources of mercury besides avoiding a computation problem concerning the edges of the CWT where the density of trajectories are low. It is irrelevant the factor of CWT be different in different sites since it will not constrain the concentration density on a second site since concentrations are weighted by the frequency of air mass.

Lines 316-342. It would be useful to include a plot of the annual Hg emissions from central Europe and discuss the changes in anthropogenic emissions alongside with changes in TGM.

Response: See response to the first comment.

Line 348-349. "We compared the regional patterns of TGM with other pollutants (CO, CO2, CH4, O3, CHCl3, CCl4, and CFCs) also measured at Mace Head and find that TGM shows a similar pattern concerning source location as the other species closely related to anthropogenic sources." Are these results shown in the paper? Have you analyzed the trends for these co-pollutants and are they decreasing?

Response: The regional pattern of TGM is displayed in Figures 2 and 4. The regional patterns for the other mentioned species are not showed in this paper since they are well represented in the literature, included by satellite data products. All species considered in the factorization, which have trended in time-series, upward (CH4, N2O) or downward (CFCs), are detrended in the factorization.

Lines 352-367.  This discussion of the CWT results and Figure 4 were discussed in the previous section.  This discussion should be combined with the previous section.

Response: The discussion of Figure 4 was moved to 3.2 as proposed by the reviewer. Thanks for the comment.

Line 376-377.  "The source apportionment indicates a baseline factor characterized by high load of anthropogenic species accounting for 65% of TGM mass."  Has the percentage change over time?  Was the PMF analysis conducted on the yearly data or the entire 25 year period?

Response: The PMF was conducted on daily data. The change over time based on a monthly average is displayed in Figure 5 and discussed in section 3.3. A new statement improving the description of PMF is presented in the new version of the article.

Line 383.  It is not clear from the discussion how the baseline factor was extracted from the dataset.  What pollutant markers were used to assign a PMF factor to the baseline factor? Please show a plot of the factor profiles from the PMF analysis.

Response: An additional plot with the "fingerprint" for the factors obtained from the PMF solution are presented in the supplement of the article. Thanks for the comment.

Line 390.  What was the magnitude of the trend for the combustion factor and was it greater or smaller compared with the baseline factor?

Response: The magnitude of the trend for the combustion and ocean factors are displayed in Figure 5, together with the trend for the baseline factor.

The lack of a clear definition of the baseline factor is an issue especially if the objective was to evaluate the effectiveness of mercury control measures.  The focus should be on understanding the different anthropogenic source sectors contributing to TGM and whether their contributions have changed over time, and if there are emerging sources that we need to be aware of.  Also, what is the role of re-emissions of previously deposited Hg?  Are the source apportionment tools able to improve the understanding of Hg re-emissions?

Response: We agree with the reviewer that a factorization solving the anthropogenic and emerging sources would be very valuable. However, we are not quite there yet. The prospect of improving our source apportionment method is to use more source markers. The propagation of the eigenvector from axis rotations can develop and permit better source apportionment. However, more source markers are still needed considered.

References:

Custodio, D., Ebinghaus, R., Spain, T. G., and Bieser, J., 2020. Source apportionment of atmospheric mercury in the remote marine atmosphere: Mace Head GAW station, Irish western coast, Atmos. Chem. Phys., 20, 7929–7939, https://doi.org/10.5194/acp-20-7929-2020

Lyman, S. N., Cheng, I., Gratz, L. E., Weiss-Penzias, P., and Zhang, L., 2020. An updated review of atmospheric mercury. Science of the Total Environment, 707, 135575, https://doi.org/10.1016/j.scitotenv.2019.135575

Downloaded from STM Journals.com

---

## Author Comment (AC2)

Review ACP Custodio et al

Response to Anonymous Referee #2

Overall, it is not clear to me the purpose of this paper and how it is providing new information. The datasets presented are rich and have the potential to provide interesting new results, but as of now there is little in the paper that is new. A lot of previous work related to what is presented is not discussed. I elaborate on the main issues first and then present line-by-line comments.

Response: A response line-by-line for each comment from the reviewer is presented in the line-by-line`s comments.

**Major and General Comments**

**Discussion of the Tekran instrumentation and uncertainty should be improved, related to whether TGM or GEM is being measured.**
Line 134: "usually" – which of these stations' instruments have this filter operating and which do not?

Response: A new statement is presented with more information about the instrument used in each station. Thanks for the comment.

Line 137: "probably" – what does this mean quantitatively? Slemr et al. (2016) describes the CARIBIC aircraft measurements in the upper troposphere/lower stratosphere which are specifically in a low-humidity but high ozone environment. How do the conclusions regarding GOM compound collection from Slemr et al. (2016) apply to the surface stations, especially coastal ones, in this study?

Response: *The word "probably" is related to the process of the GOM reduction: does it happen already when adsorbed on the gold surface or during the thermodesorption. There is no experimental evidence in favour of one of these processes but the reduction during the thermodesorption is more likely. Slemr et al. (2016) concluded that Tekran measures TGM provided that GOM gets to the gold cartridges. This is a general conclusion, independent of the conditions.*

Line 139-140: "soda-lime filter.. is known to capture GOM" – please add citations here. Gustin et al. (2021) state: "The Global Mercury Observation System standard operating procedure states that a soda lime trap in front of the Tekran 2537 removes GOM, though this has not been adequately tested."

*Done*

In particular, it is not actually known whether >95% of TGM is GEM outside of polar depletion events (the discussion at the end of the sampling sites part of Section 2, lines 143-146). Please add a reference to at least one of the papers and reviews discussing the major uncertainties measuring gaseous oxidized mercury (e.g., Maruszak et al., 2017; Lyman et al., 2020; Gustin et al., 2015, etc.) as it relates to uncertainties in the operationally-defined TGM Tekran measurements. I find that Lyman et al. 2020 summarize it better than I could myself and quote from them here: "it remains unclear whether Tekran 2537 and similar mercury analyzers without upstream processing equipment measure total gas-phase mercury or only elemental mercury. Thus, these measurements are also operationally defined. We use the acronym GEM (gas-phase elemental mercury) to describe them, though they likely include GEM and some portion of $Hg^{II}$ compounds that exist in the atmosphere, and the amount of $Hg^{II}$ they include likely depends on the sampling configuration and the chemical and physical conditions of the atmosphere. Also, when KCl denuders are used upstream of elemental mercury analyzers, some of the captured $Hg^{II}$ is reduced to elemental mercury and measured in that form (Lyman et al., 2010). Some have used the acronym TGM (total gaseous mercury), rather than GEM, but, to our understanding, no information exists about the percentage of gas-phase $Hg^{II}$ that is analyzed by elemental mercury analyzers with an upsteam KCl denuder or without upstream sample processing. Many have asserted that this issue is inconsequential, because atmospheric $Hg^{II}$ concentrations are low relative to $Hg^0$ (Ci et al., 2011; Fu et al., 2012a), but this assertion has been shown to be inaccurate in some environments (Fu et al., 2015; Obrist et al., 2011; Swartzendruber et al., 2006; Weiss-Penzias et al., 2009), especially when the low bias in KCl-denuder based GOM measurements is taken into account (Huang et al., 2013; Lyman et al., 2016)."

Response: *The discussion of the TGM/GEM issue has been rewritten.*

Overall, I am confused when the methods state that it is concluded only GEM is being measured but then all data is treated as TGM. Should it be GEM and not TGM? Interpreting the observations as GEM is consistent with other studies (e.g., Travnikov et al., 2017) but a more thorough discussion of the caveats is warranted.

Response: *Changed to GEM throughout the manuscript.*

**Clarify and provide more information on the methods used.** While I understand not wanting to repeat too much from previous work, as the sub-sections "Back-Trajectory Analysis" and "Concentration-weighted trajectories (CWT) and probability mass function (PMF) models" are written now the reader cannot understand what is being done. More information must be provided so that what is in the current paper becomes sufficient, and only the reader who wants to learn more specific details needs to look to Custodio et al. (2020).

Back Trajectory analysis: I found this section very confusing. Perhaps a schematic (potentially in the supporting information) could be added showing each step of this method. Please provide more information on when the trajectories are initiated. Is it every 12 hours for all days and years of data or a single year of data? If it is a single year, which specific year is it? Please explain and justify all choices made for the HYSPLIT simulations carefully including the arrival times of

midnight and 12pm (is this local time or UTC?) and arrival heights of 50 and 500m. Combining the concentration-weighted trajectories for all 5 sites, rather than analyzing them on a site-by-site basis, seems like it waters down the results and adds uncertainty especially as the number of sites with data is different for different years.

Concentration-weighted trajectories are described in the "Back-Trajectory Analysis" section, yet they are in the title of the following section.

Line 169: I am much more familiar with PMF standing for the EPA positive matrix factorization method. In Custodio et al. (2020), PMF is used as an abbreviation for positive matrix factorization, not "probability mass function".

CWT and PMF section: these two paragraphs are quite confusing with some very long sentences that are difficult to follow.

The methods in Section 3.2 are also in need of additional explanation, including "Kernelregression", "LSQF" (including defining this acronym in the main text), how the annual cycle was removed from Figure 3, and how confidence intervals were calculated (or what the trend ± values actually mean: are the ± values the standard deviation, standard error, 95% confidence interval from bootstrapping?). It is not clear from the description here or in Figure 3 that the regression is being performed on monthly medians. The standard deviation or confidence intervals reported for trends in this paper are much narrower than in previous studies cited (e.g., Cole et al., 2013). Why is this, and how do the methods in calculating the confidence intervals in Cole et al. (2013) differ from this study?

Response: We appreciate all these comments. A few of them are also highlighted in the RC1 as the description of the HYSPLIT features. As we replied to RC1, the description of that section was reworded. The reasons for combining trajectories from 6 sites and the possible uncertainties associated with it are also presented in response to the RC1. Many other comments highlighted here by RC2 are repeated in the line-by-line comments where the authors address a precise response.

*Moreover, there are several differences in the trend calculation by Cole et al. (2013) and those in Table 1: Cole et al. (2013) used daily averages and non-parametric Man-Kendall test. We have chosen a simpler method of LSQF of monthly medians because in this table we focus on comparison of the individual sites with Mace Head. As mentioned in the revised text, the use of monthly medians instead of daily averages reduces substantially the influence of depletion events at polar stations and of pollution events at midlatitude ones. In addition, confidence intervals provided by Man-Kendall test are larger than those provided by LSQF for the same confidence level (95%).*

**Improve statistical analysis of the observations and distinguish what is new in this paper, including relative to Custodio et al. (2020).**

In the opening section of the results (Lines 194 – 215), please provide quantitative metrics of the skewness and kurtosis of the distributions, and differences between sites or average vs. median rather than qualitative descriptions.

Response:  *The authors have no idea about the information which could be gained from the skewness and curtosis – (they are not necessary on our opinion) the discussion has been slightly edited.*

Can you quantitatively define pollution events, such that it is clear that "Events with local and regional pollution are also missing at Andøya"? I also am confused about how Weigelt et al. (2013) is a reference for "frequent events with local and regional pollution" at Waldhof. As far as I can tell they mention a single time that a local combustion plume was observed and overall put Waldhof on par with other rural background sites. Again, a quantitative definition of pollution events would be helpful here.

Response: I am afraid of not taking mean, median, maximum, minimum and distribution as not a quantitative metric. In addition, it is not clear for the authors which information could be gained from the skewness and curtosis.

*As the paper is focused on PMF and factorization analysis, only an overview of the measurements at the individual sites are given to characterize them and to show their different characteristics. A detailed analysis (frequency and severity of the depletion or pollution events, trends, etc.) of the data from individual sites has been mostly published in papers listed in the section "References". To repeat this analysis here would only lengthen the paper.*

The 3.1 seasonal variation section does not seem to add any new information as seasonal patterns at these sites have been published in other studies previously. What would be new is if you could add more detailed and rigorous comparison of the seasonality between sites (they are "similar") or from the longer time record if the seasonal cycle has been changing over time at any individual site. An example of a more detailed analysis of observed seasonality at Arctic sites including statistical tests of the significance of differences between seasons and between sites' seasonality is presented in Angot et al. (2016). The amplitude of the annual cycle at each site is only referenced in the caption of Figure 3. This should be in the main text along with an explanation of its calculation. In section 3.2, the comparison of the trend at each site to Mace Head because it is the "baseline trend" is not explained. Table 1 is confusing as it currently stands, please add more explanation in the text and in its caption. Please also explain why and how the different time periods of trends are calculated for Mace Head. Similarly in this section, the comparison of trends between sites are described qualitatively (e.g., "similar", "somewhat higher", "substantially higher") rather than quantitatively.

The paper would be stronger if the PMF analysis was performed at other sites besides Mace Head, where it has been done before for a subset of the time period presented.

Response: Wang and Mao (2021) already reported a detailed and recent analysis of seasonality in the Arctic. The amplitude of the annual cycle obtained for each site and the deployed time series decomposition method will the added in the new version of the manuscript. A clearer statement about the baseline factor is presented, and more information about PMF features and the solution.

**The conclusions drawn in the paper do not seem to be supported by the analysis.** The discussion in Lines 281-288 is speculative. I do not have enough information to evaluate this argument.

- While the back-trajectory analysis may show transport from Greenland, how do we know from observations that the atmospheric mercury there was elevated?
    - Response: The Villum station is located in Greenland.
- I would need to see a timeseries of the AO and NAO plotted against the mercury concentrations in order to evaluate how extreme the AO was for those 2 particular years and its relationship to mercury concentrations over the full time period.
- Response: AO and NAO data (time series) are freely available online (NOAA, NCAR, Met Office, data center stored) and can be provided in response to this comment as plotted below.

[Figure]

- Hawkings et al. (2021) is reporting measurements from 2018 as far as I can tell. How is this related to 2010 and 2013?
- Response: Hawkings et al. (2021) report a phenomenon, a natural source of mercury in Greenland

The discussion in lines 316 – 334 referencing specific policy actions and their relationship to observed trends is lacking evidence. Some specifics:

1) Minamata convention signing in 2013: The assumption that policies were already implemented at this point before the treaty went into force, and that this would be observable at Mace Head, ignores previous work done showing the difficulty in linking observations to the Minamata Convention and other policies (e.g., Selin, 2014; Giang et al., 2018). Whether emissions may have already changed at the timing of the treaty signing could be investigated directly by examining emissions inventories over this time period (e.g., Streets et al., 2019 global inventory for 2010-2015; GMA2018 global inventory for year 2015 (AMAP/UNEP 2019); Leclerc et al., 2019 – 2000 – 2014 for the EU;)

2) air pollution policy in the United States: I assume this is referencing what eventually became the MATS rule (https://www.epa.gov/mats) which was tied up in the courts for a number of years (and potentially still is as far as I can tell). Zhang et al. (2016), while the inventory itself is only presented until 2010, did analyze changes in Hg emissions from US coal-fired power plants from 2005 to 2015 (see Figure 1) and attributed a significant portion of the decline to mercury-specific control technologies resulting from the planned MATS rule coming into force. However, after 2016, when the MATS rule became tied up in the courts, I would not have expected coal-fired utilities to continue operating expensive mercury-specific controls if they did not need to. It would be possible to investigate this by applying similar techniques of Zhang et al. (2016) to later years of data for US power plants.

3) The 2007 policy banning mercury in non-electrical measuring devices in the EU could be quantified based on existing inventories (e.g., Horowitz et al., 2014; Streets et al., 2017, 2019; GMA from AMAP/UNEP 2013 or AMAP/UNEP 2019 reports) rather than speculated. From these studies, this sector is not a large portion of the mercury emissions and thus I doubt it would be able to be observable in atmospheric trends.

The paper discusses downward trends and states them as such in the conclusions in lines 426 – 427, but at four out of the six sites the trends shown in percentage form are not statistically significant (confidence interval includes 0% yr$^{-1}$). In Table 1 the trends are all statistically significant – how is this the case? As I mentioned earlier, the standard deviations(?) or confidence intervals of the trends shown in Table 1 are much narrower than in previous studies like Cole et al. (2013). Much past work has discussed how Arctic sites do not have a decreasing trend unlike mid-latitude sites. Skov et al. (2017) performed a more detailed statistical analysis at Villum from 1999 – 2017, finding no significant annual trend in GEM, and their comparison with a model allowed for more convincing attribution of emissions to the mercury concentrations observed.

The "seemingly non-monotonic downward trends with inter-annual ups and downs are not well-explained" also ignores previously published work. The attribution of trends to anthropogenic emissions is not convincing. Other potential mechanisms are not discussed and natural sources and sinks are referenced only briefly. A detailed review of observed trends and potential mechanisms including anthropogenic emissions, natural emissions/re-emissions, sinks, and oxidative capacity is also provided in Lyman et al. (2020) and references therein. Declines in anthropogenic emissions are referencing studies who only estimated anthropogenic emissions until 2008 (Streets et al., 2011) or 2010 (Horowitz et al., 2014; Zhang et al., 2016). There is no reference to a more recent study (Streets et al., 2019) that estimated mercury emissions as well as modeled their impact on concentrations from 2010 to 2015, including discussions on a regional basis. Another recent study that is not referenced here (Wang and Mao, 2021) specifically examined how trends in emissions from the EDGARv4.tox2 inventory (1970 through year 2012; Muntean et al., 2018) and the location of Arctic fronts interacted to affect transport of emissions to the Arctic. Finally, interannual variability in Arctic atmospheric mercury from 1979 to 2008 was also investigated in Fisher et al. (2012) and was related to specific environmental and climatic factors.

Response: The comments on this section are very important, and I would like to put light on them. First of all, we want to state that the significance in the trend here reported is significant, on both methods used to calculate it. The standard deviation here presented are much smaller than std reported in the literature because this is the first study decomposing the time variation. The others studies do not modulate seasonality, and they do not compute annual variability in the trend uncertainties.

As reported by Lyman et al. 2020, the trend in the atmospheric mercury concentration can be explained only by emission. However, inventory is not the only metric for emission. In addition, emission inventories have large uncertainties up to 60%. As explained in the response of RC1, there are many inconsistencies in the inventory. In this study, we calculate and discuss emission from the perspective of receptor modelling. More detail explaining this issue was reported on the first response to the RC1.

**Figures are not adequately described prior to when interpretations based off them are given.**

- Figure 2: interpretation is given in Lines 218 – 220, but it is not until Line 237-238 that it is explained what is specifically in the Figure. This explanation is also not linked to the methods section where more detail is given on the density map. However, there is still not enough information – for example, what is the size of the grid cells in Figure 2? how was this chosen?

- Response: The Figures as the representation of the data are discussed throughout the manuscript. The spatial resolution of the density map is constrained by trajectory backward time with a number of segments equal to 84. We believe that the information provided about the figures shown in the manuscript would be enough for anyone to reproduce it. I understand and appreciate the comment; however, I am not sure if a full description of the CWT function, already reported in the literature, will help to understand the manuscript.

- Figure 4: Interpretations of Figure 4 are given in Lines 282, 297-299, 302-306, and 312313, but the figure is introduced / described in Lines 352-353. Please move the description of Figure 4 from Lines 352-353 before any interpretations of the Figure are given.
    - Response: The sentence was reworded. Thanks

**Line-by-line comments**

line 31: is there only a single probabilistic source contribution factor?
Response: Sentence reworded.
Line 35: Is this superscript 1 a citation or footnote? Please clarify whether these are results you have found or you are citing other results.
Response: The superscript -1 (year$^{-1}$) stands for per year.
Line 38: I would appreciate if the "other atmospheric trace gases" are listed here or at least a few examples are given.
Response: The trace gases used to marker sources are presented in the next section.

Lines 37-39: This sentence is confusingly worded. Please clarify and potentially split into multiple shorter sentences.

Response: The sentence was reworded based on comments of RC1.

Line 40: Instead of "accessed" I believe you mean to say "assessed"

Response: Sentence reworded. Thanks

Line 42, Line 68: I similarly would like to know what the long-lived anthropogenic species are, or at least a few examples.

Response: A new statement is presented throughout the manuscript specifying the trace gases considered in this study.

Line 71: In this discussion of regional and hemispheric trends, the mention of global trends doesn't seem to follow. Is there a reference that hypothesizes or inventory that estimates that global anthropogenic mercury emissions are decreasing? Perhaps this should be replaced by "hemispheric" instead.

Response: *According to AMAP/UNEP 2013 and 2019 the emissions are increasing since 1995.*

Lines 72-73: "still not unknown sources" is unclear. I thought it would make more sense to say "still unknown sources", but how could one do a time series analysis of unknown sources? Overall this sentence is confusing.

Response: We agree. The sentence was reworded.

Line 80: "On this raw" – I do not know what this means.

Response: The sentence was reworded.

Line 81: "baseline factor" – the meaning of this is not explained. Perhaps this should not be mentioned in the introduction, but explained instead in the results and also methods. Otherwise, please explain what this means in a way that does not require reading the methods first.

Response: That is true. Thanks for the comment. Sentence reworded.

Line 84-85: I would add "observed" in front of "atmospheric mercury temporal variability" to distinguish it from models that you are trying to constrain. This sentence is also a bit vague – what kind of temporal variability (seasonal? interannual? or other analysis) and what kind of cycling predictions by the model? (e.g., concentrations, deposition?). I have not again seen a reference to constraining models outside of the introduction.

Response: Thanks for the comment. The suggestion was taken on the paragraph or in another part of the manuscript.

Section 2 – Sampling Sites: A map with the stations labeled would be helpful. Although the latitude and longitude are given in individual paragraphs, having this information in a table or on the map would be helpful so it is all in one place. Sometimes the altitude of the stations is not given. Similarly the altitudes could be included in a table or map label.

Response: Labels identifying the station location was added in Figure 2. Height for all stations was added in the new version.

Line 96-97: At this point, we do not have the latitudes of the stations and it would be helpful to have seen these (or the locations on a map) for the comparison of Andøya and Amderma. Is it known from previous analyses or your own analysis that the mercury concentrations at Andøya are more similar to those observed at other mid-latitude sites? (Angot et al., 2016 found this, for

example, in their analysis of seasonality of Arctic sites). It seems your own analysis from the seasonality in your Figure 2 also supports this, but at this point in the paper the classification of Arctic vs. Midlatitude sites is not well-motivated. Explain why this classification will be useful.

Response: The seasonality composed an important aspect of the trend evaluation since the time series were decomposed (or detrended).

*In addition, *Reference Angot et al. (2016) added in the manuscript*.

Line 109, 115, 120: Altitudes are not given for the Waldhof, Villum, or Andøya sites.

*Done*

Line 120-122: To me, it does not add anything to mention ALOMAR here.

Response: Sentence reworded.

Line 130: At the end of every paragraph introducing the measurement sites, I was wondering what instrument they were operating. It would be helpful to give the reader a heads up (perhaps at the end of the first paragraph) that all sites operated Tekran instruments and more detail will be given at the end of the section.

Response: Done. Thanks for the comment. More information on the instrument is given following comments of RC1.

Line 152: write out the acronym for HYSPLIT the first time it is used. 153: make clear that Stein et al. (2015) is the reference for HYSPLIT model. As it is written now it sounds like Stein et al. (2015) describes the method of choosing the trajectory heights and time initiations, but these are choices made by the authors here which must be explained.

Response: Thanks for the comment. The sentence was reworded.

Line 171-173: this sentence should be rewritten and split into multiple sentences for clarity. Right now it seems like some words are missing for it to make sense.

Response: The  sentence was reworded.

Lines 179-181: this is also a very long and hard to understand sentence that should be split up for clarity.

Response: The sentence was reworded and relocated in the manuscript based on comment of RC1.

Lines 188 – 192: rather than list the number of days of data separately from the station names, include this information in a table. The number of months are listed in Table 1. What information does the number of days add? is the data averaged daily? Are daily averages what is summarized in Figure 1?  Figure 1. please state in the caption the temporal frequency of the measurements presented (e.g., daily averages?).

Response: The summary of the dataset presented in Figure 1 has hourly time resolution (this information was added to the Figure label. The periods of data availability are important and any trend evaluation and comparability. We report the time availability of the data in the unity of days. Later on, the daily average will be used in the factorization to assess factors contribution.

Line 195 – 199: This sentence is also very long and should be split into multiple sentences for clarity. Also, what is the detection limit of the Tekran instruments at these sites? Are minima of 0.0 vs. 0.1 ng m$^{-3}$ real and different? It would be more useful to know the number of depletion events in each year rather than just the descriptor "frequently". Referencing studies of depletion

events in and around these particular sites should happen sooner in this paragraph. A detailed analysis of changes in the frequency and timing of depletion events as well as comparison with previous studies is presented in Angot et al. (2016), referencing in addition to the studies already cited here, Berg et al. (2013) and Chen et al. (2015).

Response: Yes, 0.0 and 0.1 ng m-3 are different. The limit detection of Tekran is 0.04 ng m-3. The statistics of the depletion events are displayed in Figure 1. I am afraid that a non-scholastic reference of the number of depletion events will not help in the modulation of the mercury trend in the Arctic. In addition, a detailed analysis of changes in meteorology and mercury mobility was recently presented by Wang and Mao (2021).

Line 202 – 203: I am confused how the altitude of the Zeppelin station is relevant here as the area around the station and higher-altitude areas near it can also be snow-covered.

Response: Yes, The elevation of the Zeppelin station explains quite a lot of its differences compared to the others Artic station. At that latitude, the boundary layer is very shallow. Zeppelin station can be quite frequently above the boundary layer.

Line 261: What does "regional emissions" mean? Is this anthropogenic emissions? Natural/legacy re-emissions from land?

Response: This is an excellent question. Certain it could be linked to all components that the reviewer highlight. However, the sentence states the higher level of mercury at Waldhof, a rural background site much closer to mercury anthropogenic sources than any of the others stations. The presence of regional sources is displayed in the higher central tendency in the TGM concentration (average) and a particularly tail to the right in the concentration distribution, characterizing what we call "statistic of extremes".

Line 279-280: I do not understand the meaning of this sentence and how it relates to the previous sentence. Changing the discussion from qualitative to quantitative might help.

Response: Sentence reworded.

Line 289-291: Please add a citation for this sentence.

Response: The subsequent citation in the manuscript paragraph contemplates the aforementioned sentence.

Line 298-299: Please show and cite specific anthropogenic emission estimates for this region, otherwise this sounds like speculation.

Response: To be coherent with our source apportionment narrative presented next section, we call it anthropogenic emission. The specification of the anthropic emission would request a source apportion more elaborated than CWT or a PMF with more source marks to solve and address the strength of different anthropogenic sources.

Line 300: Bringing up uncertainties in the HYSPLIT modeling approach is critical and there is not enough explanation of what you mean here. I do not understand the shift to the South that is being described.

Response: As a Lagrangian method, HYSPLIT explicitly calculates the standard deviation of the horizontal wind speed as a measure at a Gaussian distribution. The receptor modelling estimates the trajectories included in their calculation accounting the turbulent component that increases as the particle go further back in time. As backward time passes, each lagrangian particle gets

further apart from each other, creating a plume shaped concentration. You can call it the "noise" function or dispersion/turbulent component that particularly affect the edge of our estimation as observed for South Europe.

Line 301-302: Please add a specific citation showing the "geolocation of TGM sources in Europe." A figure showing the locations of major point sources or gridded anthropogenic Hg emissions inventory for the region would be helpful.

Response: The sentence was reworded citing Panagos et al. (2021) https://doi.org/10.1016/j.envres.2021.111556

Line 303-304: The explanation given in Lines 281-288 seems to conflict with what is shown and described in Figure 4 here. The earlier explanation seems to be specific to 2013 and potentially 2010-2014, but similarly high levels are shown in Figure 4 over Canada and Greenland for 19972000 and 2005. Please clarify.

Response: At lines 281-288, we discuss the trend observed at Villum, its refer to results presented in Figure 3. The statement on lines 303-302 (old version) refers to the discussion of back trajectories as indicated at the beginning of the paragraph. The analysis of air mass allowed us to assess a potential evasion of mercury in Greenland in a period not covered by Villum`s data. The sentence was reworded.

Line 307: This is the only time sinks are mentioned as affecting the concentration of mercury in the atmosphere. This should be expanded in more detail.

Response: We agree. In some sense, all the sequence of the manuscript expands the discussion of changes in the sources. In the next section, we reconstruct the time series of sources solved by PMF.

Line 311: "This phenomenon can be explained only by reductions in global atmospheric mercury sources" : I am not yet convinced. Evidence must be presented to back up this claim. More discussion of specific anthropogenic emissions inventories and estimates of emissions from natural sources, legacy re-emissions, as well as sinks, chemistry, etc. is needed.

Response: This is an interesting discussion. Trends in the concentration of atmospheric mercury can be explained only by changes in emission (Lyman et al. 2020). Emission (or reemission) is necessarily associated with the "sources". The sentence was reworded.

Lines 310-311 and 312-313: A quantitative assessment of the trends in the concentrationweighted trajectory would be more helpful than qualitatively referring to the colors on the figure.

Response: A quantitative assessment of the trends at each site is presented in Table 1, Figure 3 and discussed throughout section 3.2.

Lines 314-315: The trend at Villum seems very different from the that at Zeppelin. What are they affected by if not anthropogenic emissions from continental Europe? Please reference here prior work that has been done in this space (e.g., Fisher et al., 2012; Skov et al., 2017; Angot et al., 2016; Wang and Mao, 2021).

Response: Zeppelin Station is at 474 m height at a latitude of a shallow boundary layer. This station has a very different transport dynamic compared to Villum. In addition, local sources as recently reported by Hawkings et al. (2021).

Line 321-322: Landfills are not a significant source of mercury. Horowitz et al. (2014) reviewed measurements of emissions of mercury from landfills and found they can be treated as a

longterm sink of mercury on centuries-long timescales. as their emissions of mercury are so small the lifetime is on the order of 20,000 years.

Response: We appreciate the comment. The sentence was reworded.

Line 341-342: This may be true, but there have been recent studies on other natural processes, e.g., Hg uptake to land (Zhou et al., 2021; Jiskra et al., 2018) driving seasonality and trends.

Response: We agree. In the following section, we reconstruct the ocean emission at Mace Head, and a trend was observed. Indeed, more investigation is needed to reveal the hole of natural processes in the trend.

Lines 348-350: Where is this shown?

Response: The regional pattern of GEM is displayed in Figures 2 and 4. This paper does not show the regional patterns for the other mentioned species since they are well represented in the literature, included by satellite data products.

Lines 357-358: How do you know that the North Atlantic a sink of anthropogenic pollutants? The North Atlantic is a hot spot of net evasion in Soerensen et al. (2010) and Zhang et al. (2019).

Response: As any ocean region in general, North Atlantic is well known as a sink of anthropogenic pollutants, for example, the well-known uptake of $CO_2$. The statement is not explicitly addressed to mercury. In addition, there is a growing concern about the atmospheric-sea exchange of mercury, as recently reported by Jiskra et al. (DOI: 10.1038/s41586-021-03859-8).

Line 359-361: This hypothesis seems like it would be testable by performing chemical transport modeling experiments.

Response: We agree.

Lines 364-365: I would like to see analysis separating out these two processes, transport and the concentration in transported air. This might be similar to what was done in Wang and Mao (2021).

Response: Indeed, the two processes are essential. In order to evaluate the trend, we focus on the concentration in the air mass in the CWT. The frequency of trajectories is a factor playing a significant part in the internal variability, and its decomposition can potentially bring insight into mercury transport. As well discussed by Wang and Mao (2021).

Line 368: The discussion of factors is a surprise. Please introduce the PMF analysis here before going into the factors. It might need to be a separate section.

Response: We understand how this point is important; in order to address a proper description of the model, we dedicated a whole article to present PMF. This article evaluating 25 years of atmospheric mercury concentration comes as a "part II" of Custodio et al. (2020), which was focused on the source apportionment method.

Lines 369-373: I don't understand how the trends in a factor from 2013 – 2018 can be explained by declines in emissions that were estimated only until year 2008 (Streets et al., 2011) and year 2010 (Zhang et al., 2016). Please explain.

Response: Thanks for the question. The compliance among the aforementioned references relies on their findings and statement rather than a comparison of time series.

Line 376: there is not enough information on the baseline factor given.

Response: A new statement is presented. A more detached discussion about the labelling baseline factor is discussed in the RC1 response.

Line 377: What anthropogenic species?

Response: A new statement citing the used species is presented. In addition, the chemical profile of factors is presented in the new version of the supplement.

Line 379: This is an extremely high correlation coefficient. I need to know more about what went into this factor and what we can learn from it if it is explaining 94% ($r^2$) of the variance in observed concentrations at Mace Head.

Response: Following the comments and recommendations of RC1, a new statement presenting the main features of the PMF solution are presented in the supplement material.

Line 380-381: More explanation is needed as to how Figure 4 has to do with the anthropogenic species factor.

Response: Figure 4 shows a more pronounced decrease of mercury at continental regions corroborating with PMF solution, which explains the trend by decreasing anthropogenic emission.

Lines 383 – 397: I am very confused by this section. Explaining the baseline factor in detail would help with that.

Response: Further considerations on baseline description are explained in response to the RC1 and then considered in the new manuscript version.

Line 399-400: A different reference for the residence time of mercury in the ocean would be more appropriate (e.g., Amos et al., 2014?)

Response: Indeed. Thanks for the comment

Lines 400-402: Additional references about the oceanic mercury influence on atmospheric mercury due to anthropogenic activity should be cited here like Soerensen et al. (2012); Sunderland and Mason, 2007; Sonke et al. (2018); Cossa et al. (2018); etc.

Lines 403-305: I don't think this adds much to the paper, unless you add references to studies that have examined some of these processes for mercury specifically.

Response: We reconstructed the ocean emission and presented it in our manuscript; the interpretation/evaluation of such factor time series is complex and needs further studies. At the moment, our team are focusing on the sea atmosphere interaction, and new insight into the variability of ocean emission will be given in coming studies.

Lines 410-411: This conundrum has been explained by other studies. The analysis presented here does not include any estimates of anthropogenic emissions to corroborate that the current study also shows this.

Response: Based on the receptor model approach, we estimate emission from different sources, anthropogenic, ocean, and mercury sources related to combustion processes.

Line 426-427: I think there is a typo, is the trend actually 4 ±16 % per year or should it be 4 ±1.6% per year? Also, typo Waldholf -> Waldhof

Response: Typos corrected. Thanks

Line 421-422: This is not appropriately caveated given other studies on how difficult it is to observe the impact of the Minamata Convention and other policies (e.g., Selin, 2014; Giang et al., 2018).

Response: Indeed, it is difficult and complex to evaluate the impact of the Minamata Convention and other policies. Many researchers have contributed, and here we present an evaluation of emission based on the receptor model approach.

Additional references cited in this review, not already cited by Custodio et al. (2021):
Gustin et al. (2021): https://www.mdpi.com/2073-4433/12/1/73

Lyman et al. 2020:
https://www.sciencedirect.com/science/article/pii/S0048969719355706#bb1685

Marusczak et al. 2017: https://pubs.acs.org/doi/10.1021/acs.est.6b04999

Gustin et al. 2015: https://acp.copernicus.org/articles/15/5697/2015/

Travnikov et al., 2017: https://acp.copernicus.org/articles/17/5271/2017/

Angot et al. (2016): https://acp.copernicus.org/articles/16/10735/2016/

Selin (2014): https://setac.onlinelibrary.wiley.com/doi/full/10.1002/etc.2374

Giang et al. (2018): https://pubs.rsc.org/en/content/articlehtml/2018/em/c8em00268a

Streets et al. (2019):
https://www.sciencedirect.com/science/article/pii/S1352231018308884?via%3Dihub

Leclerc et al. (2019): https://www.sciencedirect.com/science/article/pii/S0160412018330101

AMAP./UNEP 2013: https://www.amap.no/documents/download/1265/inline

Streets et al. (2017): https://pubs.acs.org/doi/abs/10.1021/acs.est.7b00451

Wang and Mao (2021):
https://www.sciencedirect.com/science/article/pii/S1352231020307603?via%3Dihub

Muntean et al. (2018): https://www.sciencedirect.com/science/article/pii/S1352231018302425

Fisher et al. (2012): https://www.nature.com/articles/ngeo1478

Berg et al. (2013): https://acp.copernicus.org/articles/13/6575/2013/

Chen et al. (2015) https://agupubs.onlinelibrary.wiley.com/doi/full/10.1002/2015GL064051

Soerensen et al. (2010): https://pubs.acs.org/doi/10.1021/es102032g

Zhang et al (2019): https://pubs.acs.org/doi/10.1021/acs.est.8b06205

Amos et al. (2014): https://pubs.acs.org/doi/abs/10.1021/es502134t

Soerensen et al. (2012):
https://agupubs.onlinelibrary.wiley.com/doi/full/10.1029/2012GL053736

 Sunderland                            and                        Mason,                         2007:
https://agupubs.onlinelibrary.wiley.com/doi/full/10.1029/2006GB002876   Sonke et al. (2018):
https://www.pnas.org/content/115/50/E11586

Cossa et al. (2018): https://bg.copernicus.org/articles/15/2309/2018/

Zhou et al. (2021): https://www.nature.com/articles/s43017-021-00146-y?proof=t+target%3D

Jiskra et al. (2018): https://www.nature.com/articles/s41561-018-0078-8

---

## Author Response (AR2)

Response to the Reviewer comments on the manuscript "Odds and ends of atmospheric mercury in Europe and over northern Atlantic Ocean: Temporal trends of 25 years of measurements".

The authors mentioned Hg emissions inventory are inaccurate and methods of estimating emissions have changed over time, and thus argued that the Hg emissions data are not suitable for trends analysis. While emissions data have uncertainties, the data are subjected to quality assurance and quality control. The emissions data are being used to inform domestic and international policies including the Minamata Convention on Hg. It is not meaningful to quantify long term trends without explaining the underlying causes. The PMF analysis does not provide enough details on sources of Hg (only three factors were identified); thus, the analysis of TGM with Hg emissions inventory is necessary.

Response: The authors are aware of the importance of emission inventories, their assurance and quality control and their relevance to international policies. Governments use emission inventories to help determine significant sources and target regulatory actions. Emissions inventories are essential to mathematical models that estimate mercury released to the environment. Inventories also can be used to raise public awareness regarding sources of pollution. Indeed, the emission of anthropogenic sectors and their change over time can be assessed with periodic updates of the emissions inventory. Methods to determine emissions are many as continuous monitoring of emissions from a source; besides the inconsistence those methods as mentioned in response to reviewer.

However, there is a conundrum in the global inventories emission trend. It does not show the downward trend observed at many long-term monitoring stations. An upward trend is observed at the emission inventory (as shown in the figure below), even while considering only the northern hemisphere.

[Figure]

Figure 5S (reviewed manuscript): Time-series of global mercury emission. Emission inventory provided by Emissions Database for Global Atmospheric Research (EDGARv4.tox2, 2018). The inventory data is available at https://edgar.jrc.ec.europa.eu/dataset_tox4#sources. *The time-series displays the time variability of 12

sectors reported as cement production (cement), combustion in residential and other combustion (comb), glass production (glass), artisanal and small scale gold production (gold_A), large scale gold production (gold_L), shipping emission (shipping), road transportation (tro-roa), chlor-alkali industry, mercury cell technology (chlor), combustion in power generation and industry (ind), and solid waste incineration and agricultural waste burning (waste).

In addition, the 2018 Global Mercury Assessment (UN, 2018) indicate that the increase of mercury emission is linked to an increase in the primary anthropogenic sector, which estimation raised up to 20%. The 2018 UNEP Report (AMAP/UNEP, 2018) presents an inventory indicating increased emissions since the '90s.

The conundrum in the global mercury emission is already reported in the literature (e.g. Zhang et al., 2016). The missed compliance among observation and emission inventory trends has been linked to the miss estimation of Hg released from commercial products and emissions from coalfired utilities after the implementation of gas emission controls. Lyman et al. (2020, and references in) reported that the observed TGM trend is not consistent with the global anthropogenic emissions inventory, in which uncertainties ranged from -33% to 60%.

As the authors stated before, compiling a global assessment based on inventories requires several assumptions and generalizations (AMAP/UNEP, 2018). Several discrepancies are observed in the mass balance-based estimation; there are large differences between estimates. The estimation itself can explain the inconsistency among the decreasing trend observed at the monitoring stations and the increased emissions from 1990 to 2015, as indicated by anthropogenic Hg emission inventories (e.g. UN, 2018, and AMAP/UNEP, 2018).

The authors understand the importance of discussing the inventories in the manuscript, even its constrained time availability. For this reason, we presented a time series of mercury emission for the northern Atlantic and Europe in the new version (Figure 5 in the new version). The emission inventory considered by the authors is reported by EDGAR, which is based primarily upon information provided by states, local, and tribal air agencies for sources in their jurisdictions and supplemented by data developed by EPA. Even constrained in time availability, the trend observed in emission inventory for the northern Atlantic and Europe is complacent with the downward trend displayed by the observation and reported in the manuscript.

A new statement is presented in the new version of the manuscript concerning this point highlighted in the comment from the reviewer.

[Figure]

Figure 5 (in the new version of the manuscript): Time-series of Europe and North Atlantic mercury emission. Emission inventory provided by Emissions Database for Global Atmospheric Research (EDGARv4.tox2, 2018). The inventory data is available at https://edgar.jrc.ec.europa.eu/dataset_tox4#sources. *The time-series displays the time variability of 12 sectors reported as cement production (cement), combustion in residential and other combustion (comb), glass production (glass), artisanal and small scale gold production (gold_A), large scale gold production (gold_L), shipping emission (shipping), road transportation (tro-roa), chloralkali industry mercury cell technology (chlor), combustion in power generation and industry (ind), and solid waste incineration and agricultural waste burning (waste).

Based on the available dataset, the PMF factors comprised a baseline, combustion and ocean/marine factor. The baseline factor refers to background, but do we know what sources are contributing to the background factor? Does this include all Hg sources, anthropogenic, natural or re-emissions? If it includes anthropogenic sources, does it include combustion sources which was resolved in a separate PMF factor? It is important to identify specific types of sources in order to inform mercury pollution control policies. The study reported a 2.7% decrease per year in the baseline factor; however, there was no explanation on what is driving the decline in the baseline factor. Similarly what is causing the increasing trend in the marine factor?

Response: The authors are afraid that most of the questions addressed in this comment, which are indeed important, cannot be answered by the PMF solution here presented. Even neither by the current possibility of receptor model technology here deployed to assess gas mercury fluxes. The authors hope that the emission information from EDGARv4 presented in the new version gives new insights on the anthropogenic sectors driving the atmospheric mercury trend down.

As presented by the authors, the significance, implication and causes of variability in the marine factor remain to be determined. It can be associated with the changing the ocean's biogeochemistry as acidification of oceans, climate change, excess nutrient inputs, or other phenomena affecting mercury ocean-air fluxes.

The authors discussed some disadvantages and limitations of receptor models like PMF in their response. Such discussion should be included in this paper considering that a major source of

atmospheric Hg is from re-emissions of previously deposited Hg, which cannot be resolved using current receptor modeling tools. Are there other parameters that can be included in the PMF model to identify Hg re-emissions (e.g. temperature)?

Response: A new statement on the limitation of the PMF solution obtained on this study is presented in the revised version "Due to a lack of source markers that could allow the propagation of the eigenvector from axis rotation to reconstruct more realistically the complexity of mercury sources, only four factors solved our factorization. However, such an approach provided be a valuable method to evaluate mercury fluxes". The re-emissions factor could be addressed only by a marker of such "source".

Given the 10-25 years of TGM data available at six monitoring sites, there needs to be a more detailed and deeper analysis of the data than the one currently presented in the paper. There were no additional analyses conducted to address this comment in the revised paper.

Response: The experimental section was reworded to attend to the information requested by reviewer 2 (it appear from lines 168 to 198). In addition, an extensive description of the experimental features is avoided since it is already reported in the literature. The authors are restricted to providing the main features of the observational sites, besides information about sampling and analytical methods. Detailed information about the six sited (as well information about the sampling in each one of them) is presented at references provided by the authors.